# Estrogen-related receptor γ causes osteoarthritis by upregulating extracellular matrix-degrading enzymes

Young-Ok Son [1], Seulki Park[1], Ji-Sun Kwak[1], Yoonkyung Won[1], Wan-Su Choi[1], Jinseol Rhee[1], Churl-Hong Chun[2], Je-Hwang Ryu[3], Don-Kyu Kim[4], Hueng-Sik Choi[4] & Jang-Soo Chun[1]

The estrogen-related receptor (ERR) family of orphan nuclear receptor is composed of ERRα, ERRβ, and ERRγ, which are known to regulate various isoform-specific functions under normal and pathophysiological conditions. Here, we investigate the involvement of ERRs in the pathogenesis of osteoarthritis (OA) in mice. Among ERR family members, ERRγ is markedly upregulated in cartilage from human OA patients and various mouse models of OA. Adenovirus-mediated overexpression of ERRγ in mouse knee joint or transgenic expression of ERRγ in cartilage leads to OA. ERRγ overexpression in chondrocytes directly upregulates matrix metalloproteinase (MMP)-3 and MMP13, which are known to play crucial roles in cartilage destruction in OA. In contrast, genetic ablation of *Esrrg* or shRNA-mediated downregulation of *Esrrg* in joint tissues abrogates experimental OA in mice. Our results collectively indicate that ERRγ is a novel catabolic regulator of OA pathogenesis.

[1] National Creative Research Initiatives Center for Osteoarthritis Pathogenesis and School of Life Sciences, Gwangju Institute of Science and Technology, Gwangju 61005, Republic of Korea. [2] Department of Orthopedic Surgery, Wonkwang University School of Medicine, Iksan 54538, Republic of Korea. [3] Research Center for Biomineralization Disorders, School of Dentistry, Chonnam National University, Gwangju 61186, Republic of Korea. [4] National Creative Research Initiatives Center for Nuclear Receptor Signals and School of Biological Sciences and Technology, Chonnam National University, Gwangju 61186, Republic of Korea. Correspondence and requests for materials should be addressed to J.-S.C. (email: jschun@gist.ac.kr)

Osteoarthritis (OA), the most common form of arthritis, is a leading cause of disability and incurs a large socio-economic cost[1]. OA is a whole-joint disease characterized by cartilage destruction, synovial inflammation, osteophyte formation, and subchondral bone sclerosis[2]. However, no effective disease-modifying therapy for OA has been developed to date. OA is caused by an anabolic/catabolic factor imbalance that can be induced by various etiologic risk factors and pathophysiological processes[3,4]. Important among the potential OA-causing mechanisms are mechanical stresses, including joint instability and injury, and factors that predispose toward OA, such as aging. These factors alter biochemical pathways in chondrocytes, resulting in degradation of the extracellular matrix (ECM)[5]. Among the matrix-degrading enzymes, matrix metalloproteinase 3 (MMP3), MMP13, and ADAMTS5 (a disintegrin-like and metallopeptidase with thrombospondin type 1 motif 5; aggrecanase-2) are known to play crucial roles in OA cartilage destruction[6–8]. Various catabolic regulators, including the

proinflammatory cytokine, interleukin (IL)-1β, can upregulate MMP3, MMP13, and ADAMTS5[9]). We previously demonstrated that hypoxia-inducible factor (HIF)-2α (encoded by *Epas1*), which is transcriptionally upregulated in chondrocytes by proinflammatory cytokines or mechanical stress, causes OA pathogenesis through upregulation of matrix-degrading enzymes[10,11]. We also recently demonstrated that the zinc-ZIP8-MTF1 axis—reflecting induction of the zinc importer ZIP8 (encoded by *Slc39a8*), zinc influx, and subsequent activation of the zinc-dependent transcription factor MTF1—acts as a crucial catabolic regulator of OA pathogenesis by upregulating matrix-degrading enzymes in joint articular chondrocytes[12,13].

OA is currently considered as a disease associated with metabolic disorders[14]. Although this is still a matter of controversy[15], a number of studies suggest an association of metabolic syndrome with OA pathogenesis[16]. Among the various regulatory molecules involved in metabolism, isoforms of estrogen-related receptors (ERRs) are currently known to regulate various metabolic

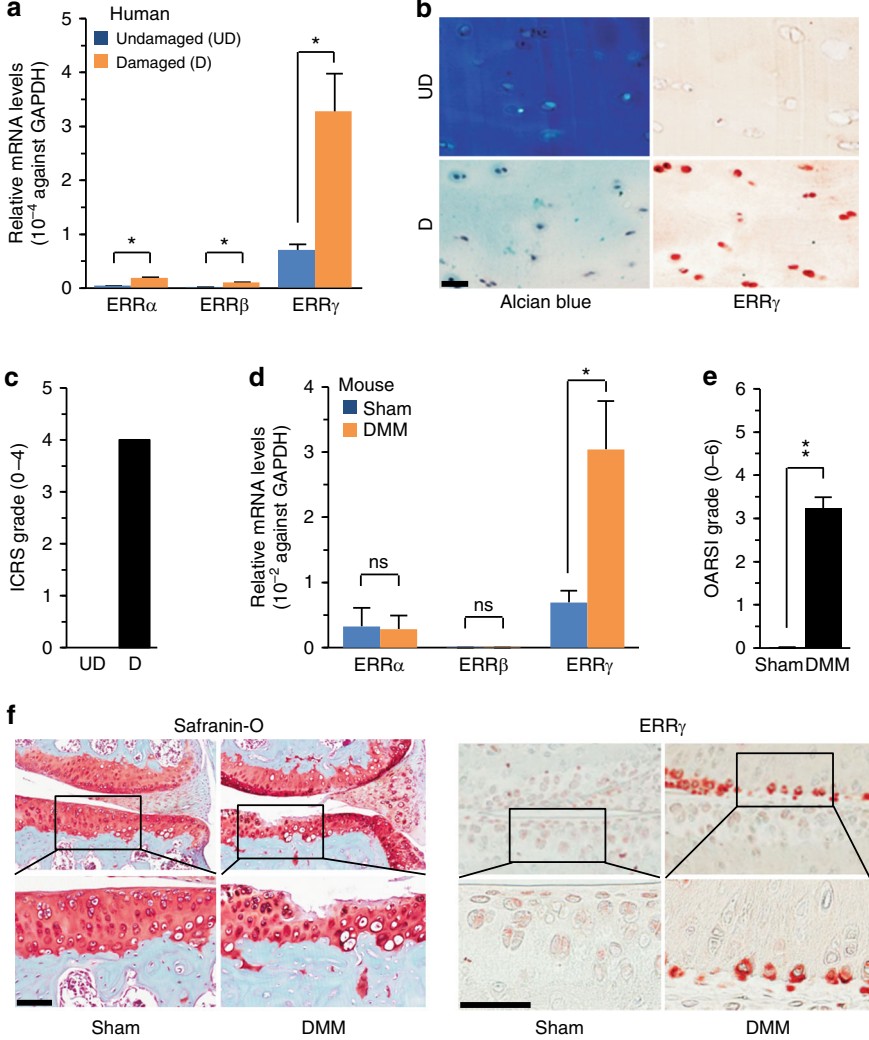

**Fig. 1** ERRγ is upregulated in OA cartilage from humans and mouse models. **a–c** qRT-PCR analysis (*n* = 11) was used to assess the mRNA expression levels of ERR isoforms in undamaged (UD) and damaged (D) regions of human OA cartilage (**a**). Representative images (*n* = 10 patients) show alcian blue staining of sulfated proteoglycans and immunostaining of ERRγ in UD and D regions of human OA cartilage (**b**). ICRS (International Cartilage Repair Society) grade of human OA cartilage sourced from individuals undergoing arthroplasty (**c**). **d–f** ERR mRNA levels were quantified by qRT-PCR (*n* = 10 mice per group) in cartilage tissues from mice subjected to sham operation or DMM surgery (**d**). Cartilage destruction was scored by OARSI grading (**e**). Representative images (*n* = 12 mice per group) of safranin-O-stained and ERRγ-immunostained OA cartilage from DMM-operated mice or cartilage from sham-operated mice (**f**). Values are presented as means ± s.e.m. (*P < 0.01 and **P < 0.001; ns, not significant). Two-tailed *t* test (**a,d**) and Mann–Whitney *U* test (**e**). Scale bar: 50 μm

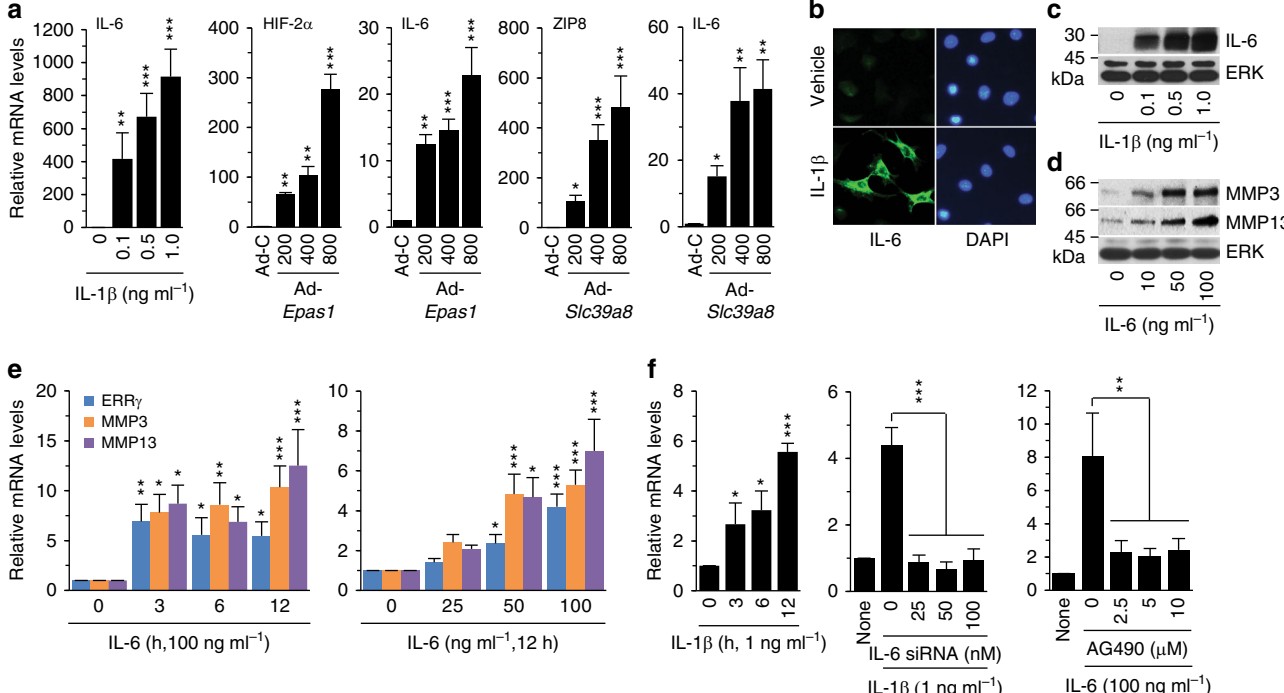

**Fig. 2** Signaling upstream of ERRγ expression. **a** qRT-PCR analysis ($n \geq 8$) of IL-6 in chondrocytes treated with IL-1β and infected with 800 MOI of control virus (Ad-C) or the indicated MOI of Ad-Epas1 or Ad-Slc3918. **b**, **c** Detection of IL-6 in chondrocytes treated with IL-1β (1 ng ml$^{-1}$) by immunofluorescence microscopy (**b**) and Western blotting (**c**). **d**, **e** Western blot (**d**) and qRT-PCR (**e**) analysis of ERRγ, MMP3, and MMP13 in chondrocytes treated with IL-6 ($n = 10$). MMP3 and MMP13 proteins in conditioned medium of chondrocyte cultures were detected by Western blotting. **f** qRT-PCR analysis ($n \geq 8$) of ERRγ mRNA levels in chondrocytes treated with IL-1β (left panel). qRT-PCR analysis ($n \geq 8$) of ERRγ mRNA levels in chondrocytes treated with IL-6 siRNA and exposed to IL-1β (middle panel). qRT-PCR analysis ($n \geq 8$) of ERRγ mRNA levels in chondrocytes treated with AG490 and exposed to IL-6 (right panel). Values are presented as means ± s.e.m. (*$P < 0.05$, **$P < 0.005$, and ***$P < 0.0005$. One-way ANOVA)

processes, including metabolic disorders and energy metabolism[17,18]. The ERR family consists of the orphan nuclear receptors, ERRα (NR3B1), ERRβ (NR3B2), and ERRγ (NR3B3). Unlike other nuclear orphan receptors, ERRs are constitutively active owing to the active conformation of their ligand-binding domain, even in the absence of a ligand[19]. Because ERRs share nearly identical DNA-binding domains, their transcriptional regulatory functions are similar to each other; thus, they regulate overlapping common target genes. For instance, ERRα and ERRγ regulate energy metabolism and promote oxidative capacity in skeletal and cardiac muscle[18,20,21]. However, individual ERRs also exhibit isoform-specific functions. For example, ERRγ regulates iron homeostasis in various tissues[22,23], ERRβ regulates embryonic stem cell pluripotency and self-renewal[24,25], and ERRα acts as a determinant of breast cancer etiology[24,25].

The roles of ERRs in cartilage biology are beginning to be investigated, with most studies focusing on the functions of ERRα[26]. For instance, studies have shown that ERRα activates the Sox9 promoter in a rat chondrogenic cell line[27] and rat mandibular condylar chondrocytes[28], and is required for cartilage development in zebra fish[29]. In contrast, cartilage-specific ERRγ transgenic (Tg) mice exhibit chondrodysplasia, in which repressed chondrocyte proliferation decreases the proliferative zone of the growth plate and reduces bone length[30]. It has also been reported that ERRα expression is reduced in bones and joints during collagen-induced inflammatory rheumatoid arthritis[31]. However, the functions of ERRs in regulating joint cartilage homeostasis and/or OA pathogenesis have not yet been studied, although an in vitro study by Bonnelye et al.[32] suggested a possible association of ERRα with OA pathogenesis. They demonstrated that IL-1β in human OA chondrocytes upregulates ERRα, which directly targets genes encoding MMP3 and SOX9.

In this study, we used human OA cartilage and mouse models of OA to investigate whether ERRs are associated with OA pathogenesis. Here, we report that ERRγ is a novel catabolic regulator of OA pathogenesis. We found that, among the ERR family members, ERRγ was specifically upregulated in cartilage from human OA patients and various mouse models of OA. Our gain-of-function (adenovirus-mediated overexpression in joint tissues or Col2a1-Esrrg Tg mice) and loss-of-function (Esrrg+/− mice or shRNA-mediated knockdown) approaches clearly indicated that ERRγ acts as a novel catabolic regulator of OA pathogenesis, at least in part, by the upregulating matrix-degrading enzymes, MMP3 and MMP13, in articular chondrocytes.

## Results

**ERRγ is upregulated in chondrocytes of OA cartilage.** To explore the possible association of ERRs with OA pathogenesis, we first examined the expression levels of ERRs in OA cartilage of human patients and various mouse models. Our quantitative reverse transcription-polymerase chain reaction (qRT-PCR) analyses revealed that ERRγ was significantly increased in OA-affected, damaged regions of human cartilage compared with undamaged regions of arthritic cartilage (Fig. 1a). Although the fold-increases in ERRα and ERRβ mRNAs were statistically significant, their expression levels were negligible relative to those of ERRγ (Fig. 1a). Consistent with the increased mRNA levels, immunostaining revealed that ERRγ protein levels were markedly elevated in chondrocytes of human OA cartilage (Fig. 1b, c). Similarly, ERRγ, but not ERRα or ERRβ, was significantly increased in cartilage from an experimental mouse model of OA induced by destabilization of the medial meniscus (DMM) surgery (Fig. 1d–f). ERRγ was also increased in the cartilage of mice

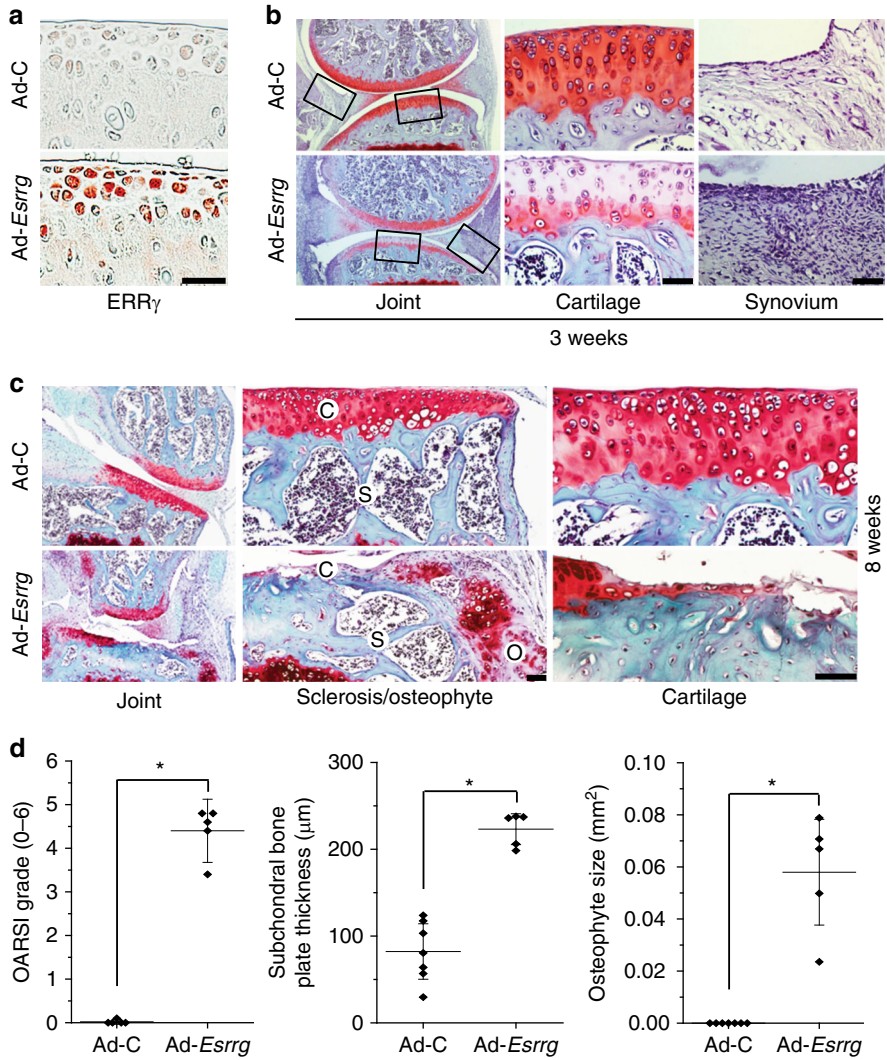

**Fig. 3** Overexpression of ERRγ in joint tissue causes OA in mice. **a**, **b** WT mice were IA-injected with Ad-C (control) or Ad-*Esrrg* to overexpress ERRγ in the joint tissues. Three weeks after the first injection, sections of mouse joint tissues were immunostained for ERRγ (**a**). Cartilage destruction and synovial inflammation were determined by safranin-O staining and H&E staining, respectively (**b**). **c**, **d** WT mice were IA-injected with Ad-C (control) or Ad-*Esrrg* to overexpress ERRγ in mouse knee joint tissues. Mice were killed 8 weeks after the first injection. Representative images of safranin-O staining, showing the whole joint (40×), subchondral bone sclerosis and osteophyte formation (200×), and cartilage destruction (400×) (**c**). OARSI grade (for cartilage destruction), subchondral bone plate thickness (for subchondral bone sclerosis), and osteophyte size were quantified (**d**; n ≥ 5 mice per group). Values are presented as means ± s.e.m. (*P < 0.005). Two-tailed *t* test and Mann–Whitney *U* test. Scale bar: 50 μm. C, cartilage; S, subchondral bone; O, osteophyte

with OA caused by the overexpression of HIF-2α via intra-articular (IA) injection of an adenovirus-expressing HIF-2α (Ad-*Epas1*)[10,11] or by IA injection of Ad-*Slc39a8* to overexpress the zinc importer, ZIP8[12,13] (Supplementary Fig. 1a, b). However, we were unable to detect ERRα in OA cartilage of humans or mice (Supplementary Fig. 2a, b). Additionally, IL-6, which acts as a catabolic regulator of OA cartilage destruction[33], increased mRNA levels of ERRγ, but not ERRα or ERRβ, in primary cultured mouse articular chondrocytes (Supplementary Fig. 2c). Because these results collectively suggested that ERRγ was associated with OA pathogenesis, we focused on the role of ERRγ in OA pathogenesis.

ERRγ expression is regulated by multiple signaling pathways, including alcohol-induced activation of the cannabinoid receptor, IL-6-mediated inflammation, fasting-induced glucagon receptor activation, feeding-induced insulin receptor activation, hypoxia, and endoplasmic reticulum stress[17]. In an attempt to identify regulators of ERRγ expression, we examined the expression of its upstream signaling components in primary cultured

chondrocytes treated with various OA-associated catabolic regulators, including IL-1β[9], HIF-2α[10,11], and ZIP8[12,13], using microarray analyses. Among the known ERRγ upstream regulators, IL-6 expression was markedly increased by IL-1β treatment and, to a lesser degree, by the overexpression of HIF-2α or ZIP8 (Supplementary Table 1). We therefore further characterized the involvement of IL-6 signaling in ERRγ expression in chondrocytes. Similar to microarray analyses, our qRT-PCR analyses indicated upregulation of ERRγ by IL-1β treatment and, to a lesser degree, by overexpression of HIF-2α or ZIP8, in primary cultured mouse articular chondrocytes (Fig. 2a). IL-1β also increased IL-6 protein levels in chondrocytes, as determined by immunofluorescence microscopy and Western blotting (Fig. 2b, c and Supplementary Fig. 9). IL-6 treatment of chondrocytes increased ERRγ mRNA levels in a concentration- and time-dependent manner (Fig. 2e). IL-6 also increased mRNA levels and secreted levels of MMP3 and MMP13 protein (Fig. 2d, e), which are encoded by target genes of IL-6 and are associated with OA cartilage destruction[33]. Additionally, IL-1β-induced

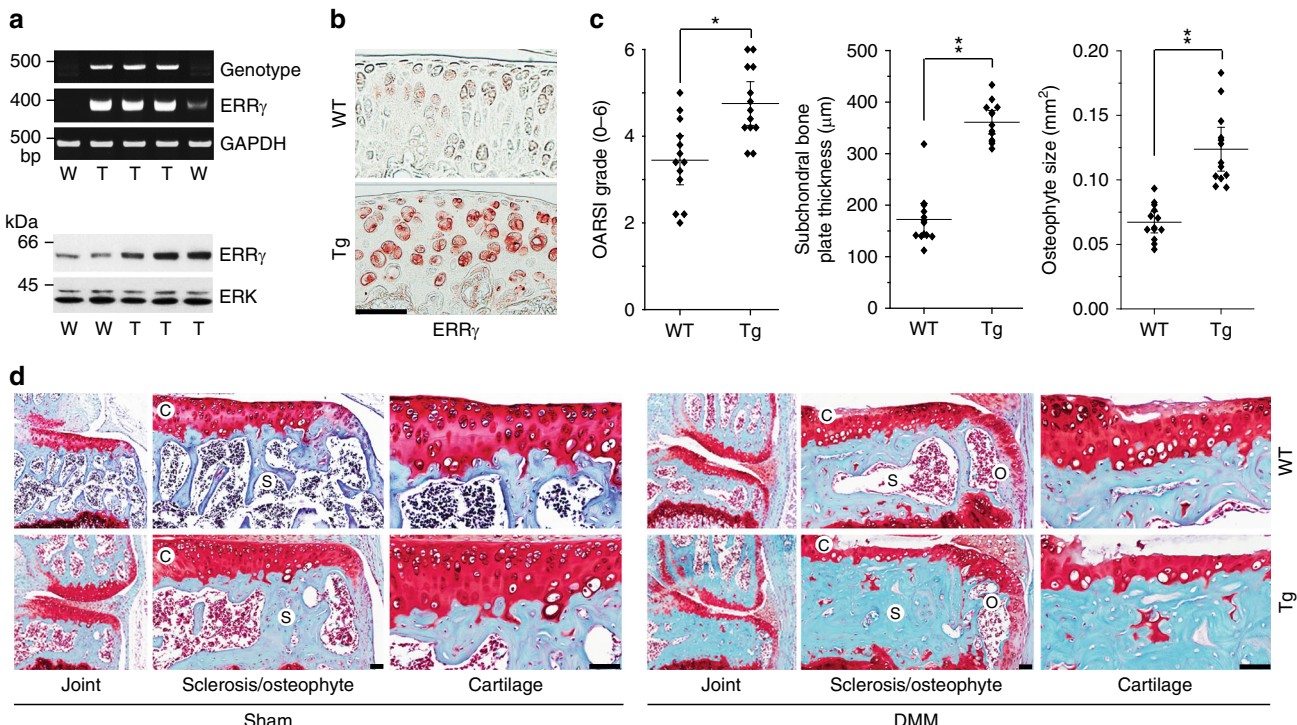

**Fig. 4** *Col2a1-Esrrg* Tg mice exhibit enhanced experimental OA. **a** Genotypes and ERRγ mRNA and protein levels were determined from primary cultured chondrocytes isolated from *Col2a1-Esrrg* Tg mice and WT littermates. GAPDH and extracellular signal-regulated kinase (ERK) were used as loading controls. **b** Immunostaining for ERRγ protein in cartilage of *Col2a1-Esrrg* Tg mice and WT littermates. **c**, **d** *Col2a1-Esrrg* Tg mice and WT littermates were subjected to sham operation or DMM surgery. OARSI grade, subchondral bone plate thickness, and osteophyte size were quantified in WT and *Col2a1-Esrrg* TG mice (*n* = 13 mice per group), mice 8 weeks after sham operation, or DMM surgery (**c**). Representative images of cartilage destruction, subchondral bone sclerosis, and osteophyte formation determined by safranin-O staining (**d**). Values are presented as means ± s.e.m. (*$P < 0.001$ and **$P < 0.0001$). Two-tailed *t* test and Mann–Whitney *U* test. Scale bar: 50 µm

upregulation of ERRγ was abrogated by knockdown of IL-6 with specific small interfering RNA (siRNA) (Fig. 2f). IL-6-mediated inflammation is known to regulate ERRγ expression via JAK/STAT signaling[17]. Consistent with this, inhibition of JAK/STAT signaling with AG490[34] abrogated IL-6-induced ERRγ expression (Fig. 2f). These results collectively indicate that the IL-1β/IL-6 axis contributes to ERRγ expression in articular chondrocytes. This is consistent with our previous work showing that IL-6 plays an essential role in OA cartilage destruction by upregulating MMP3 and MMP13[33].

**Ectopic expression of ERRγ causes OA pathogenesis**. To examine the in vivo role of ERRγ in OA pathogenesis, we ectopically expressed ERRγ in knee joint tissues of 10-week-old male mice via IA injection of an adenovirus-expressing ERRγ (Ad-*Esrrg*). We previously demonstrated that an adenovirus system effectively delivers genes to cartilage and other joint tissues[10–13]. Consistent with our prior findings, immunohistochemical staining indicated that ERRγ was effectively overexpressed in cartilage (Fig. 3a), as well as the meniscus and synovium (Supplementary Fig. 3a). ERRγ overexpression, triggered by three weekly IA injections of Ad-*Esrrg*, caused synovitis, as determined by H&E staining (Fig. 3b). At this stage, safranin-O staining indicated loss of glycosaminoglycans in articular cartilage above the tidemark, without apparent development of osteophyte or subchondral bone remodeling (Fig. 3b). However, after 8 weeks, joint tissues IA-injected with Ad-*Esrrg* showed osteophyte development and subchondral bone sclerosis, with severe erosion of cartilage (Fig. 3c, d). We also saw a loss of glycosaminoglycans in articular cartilage in both juvenile (8-week-old) and young adult (10-week-old) mice (Supplementary Fig. 3b), suggesting that developmental

stage is not a critical determinant of ERRγ-induced cartilage damage. Indeed, overexpression of HIF-2α, which was previously shown to cause OA cartilage destruction[10,11], induced cartilage damage in mice at various ages from 8 to 20 weeks old (Supplementary Fig. 3c).

IA injection of Ad-*Esrrg* caused ectopic expression of ERRγ in all joint tissues, including the synovium and cartilage (Supplementary Fig. 3a). To examine the cartilage-specific functions of ERRγ in OA pathogenesis, we generated cartilage-specific ERRγ Tg mice (*Col2a1-Esrrg*) using the *Col2a1* enhancer and promoter[10–13,35]. Primary cultured chondrocytes from *Col2a1-Esrrg* Tg mice exhibited upregulation of ERRγ at both mRNA and protein levels (Fig. 4a and Supplementary Fig. 9). ERRγ protein levels were also elevated in the cartilage and meniscus, but not synovial tissues, of *Col2a1-Esrrg* Tg mice (Fig. 4b and Supplementary Fig. 4a). Because a previously generated cartilage-specific ERRγ Tg mouse was reported to exhibit chondrodysplasia, in which repressed chondrocyte proliferation decreases the proliferative zone of the growth plate and reduces bone length[30], we characterized skeletal development in our Tg mice. Skeletal staining of E18.5 embryos revealed that skeletal structure was similar in Tg mice and wild-type (WT) littermates (Supplementary Fig. 4b). Additionally, alcian blue staining of the metatarsal bone of 2-week-old mice revealed that the lengths of resting/proliferative and hypertrophic zones of the growth plate were similar (Supplementary Fig. 4c). Thus, in contrast to the previous report[30], our data collectively indicate no significant difference in skeletal development between our Tg mice and WT littermates.

Compared with WT littermates, DMM-operated *Col2a1-Esrrg* Tg mice exhibited significantly more cartilage destruction, as

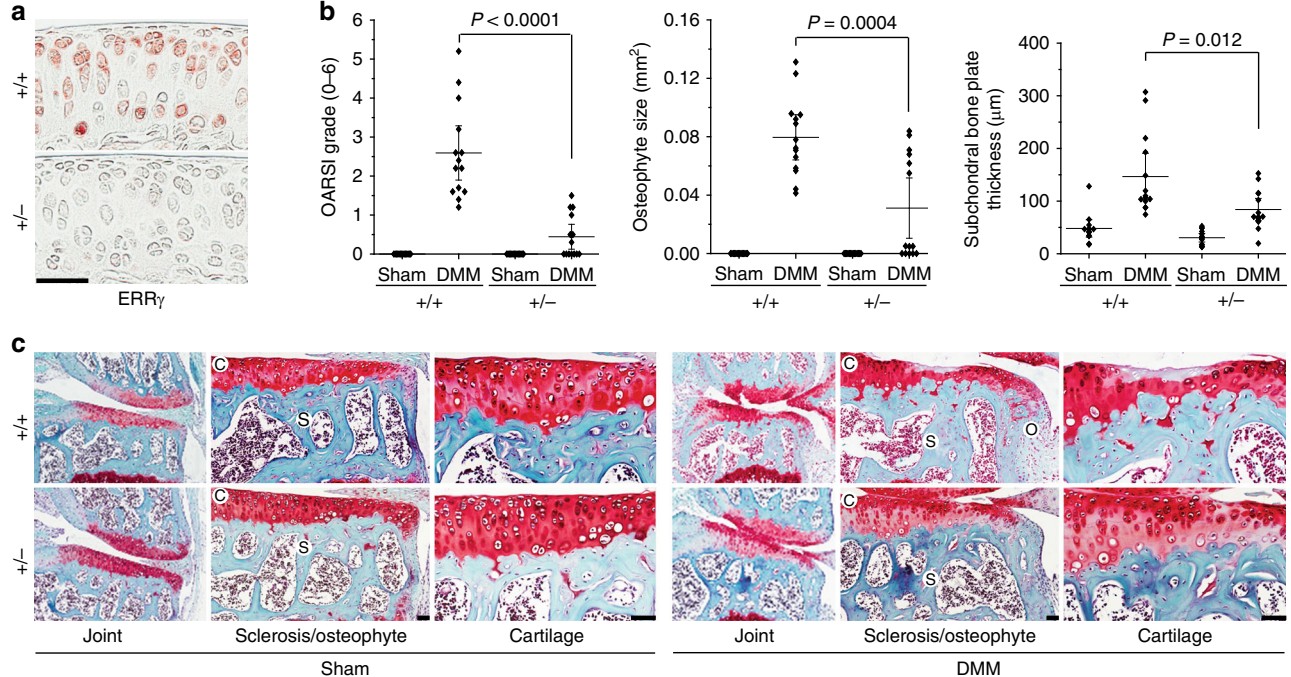

**Fig. 5** Genetic ablation of *Esrrg* abrogates OA pathogenesis in mice. **a** Immunostaining for ERRγ proteins in cartilage of *Esrrg*$^{+/-}$ and WT littermates. **b**, **c** *Esrrg*$^{+/-}$ mice and WT littermates were subjected to sham operation or DMM surgery. OARSI grade, osteophyte size, and subchondral bone plate thickness were quantified in WT and *Esrrg*$^{+/-}$ mice (*n* = 14 mice per group) (**b**). Representative images of safranin-O-stained joint sections showing the whole joint (40×), subchondral bone sclerosis and osteophyte size (200×), and cartilage (400×) (**c**). Values are presented as means ± s.e.m. Two-tailed *t* test and Mann–Whitney *U* test. Scale bar: 50 μm

determined by safranin-O staining and scoring of OARSI grade (Fig. 4c, d). Consistent with the enhanced cartilage destruction, DMM-induced upregulation of MMP3 and MMP13 proteins in damaged regions of cartilage was further increased in DMM-operated *Col2a1-Esrrg* Tg mice (Supplementary Fig. 4d). Other DMM-induced manifestations of OA, such as subchondral bone sclerosis and osteophyte formation, were also significantly enhanced in *Col2a1-Esrrg* Tg mice compared with WT littermates (Fig. 4c, d). We used 10-week-old mice for DMM surgery and examined OA manifestations at 8 weeks post surgery, because this technique was originally developed for use at this age of the mouse[36] and most DMM surgeries are performed at this age[37,38] (Supplementary Fig. 5a, b). Indeed, we found no apparent differences in DMM surgery outcomes at ages between 8 and 20 weeks (Supplementary Fig. 5c). Thus, the results of our gain-of-function studies collectively indicate that ERRγ is a catabolic regulator of OA pathogenesis.

**Genetic ablation of *Esrrg* abrogates OA pathogenesis**. To further investigate the function of ERRγ in OA pathogenesis, we used ERRγ-knockout (KO) mice as a loss-of-function approach. Because ERRγ-null mice die shortly after birth[39], we used heterozygous mice (*Esrrg*$^{+/-}$) for experimental OA studies. We previously showed that mice heterozygous for HIF-2α (*Epas1*$^{+/-}$) or ZIP8 (*Slc39a8*$^{+/-}$) exhibit significantly reduced DMM-induced OA pathogenesis[10,12]. Here, we found that *Esrrg*$^{+/-}$ mice exhibited reduced expression levels of ERRγ in joint tissues (Fig. 5a and Supplementary Fig. 6a, b) without any marked differences in skeletal staining (Supplementary Fig. 6c). Moreover, the DMM-induced manifestations of OA, including cartilage erosion, osteophyte formation, and subchondral bone sclerosis, were significantly reduced in *Esrrg*$^{+/-}$ mice (Fig. 5b, c). Consistent with this, DMM-induced upregulation of MMP3 and MMP13 proteins in damaged cartilage was markedly abrogated in *Esrrg*$^{+/-}$ mice

(Supplementary Fig. 6d). Additionally, the cartilage destruction caused by overexpression of HIF-2α or ZIP8 via IA injection of Ad-*Epas1*[10,11] or Ad-*Slc39a8*[12,13], respectively, was also significantly reduced in *Esrrg*$^{+/-}$ mice compared with WT littermates (Supplementary Fig. 6e).

The above results indicate that deletion of one allele of *Esrrg* is sufficient to reduce experimental OA in mice. Therefore, we further validated the function of ERRγ in OA pathogenesis by knocking down ERRγ in whole-joint tissues of mice via IA injection of adenovirus-expressing shRNA against *Esrrg* (Ad-sh*Esrrg*). DMM-induced upregulation of ERRγ in the chondrocytes of cartilage tissue was effectively abrogated by IA injection of Ad-sh*Esrrg* in DMM-operated mice (Fig. 6a). In our initial screening, we found that infection with Ad-sh*Esrrg* effectively downregulated ERRγ expression in primary cultured mouse articular chondrocytes. Consistent with this, knockdown of cartilage ERRγ significantly suppressed all examined OA manifestations, including cartilage erosion, subchondral bone sclerosis, and osteophyte formation, in DMM-operated mice (Fig. 6b, c). These results further support the conclusion that ERRγ functions as a catabolic regulator in mouse models of OA pathogenesis caused by DMM surgery.

**ERRγ upregulates MMP3 and MMP13 in articular chondrocytes**. Matrix-degrading enzymes produced by chondrocytes are essential effector molecules in the destruction of cartilage during OA pathogenesis[5]. Indeed, genetic studies in mice have demonstrated that MMP3, MMP13, and ADAMTS5 play crucial roles in OA cartilage destruction[6–8]. Here, we examined the effects of ERRγ on the expression of these matrix-degrading enzymes in primary cultured mouse articular chondrocytes. We found that overexpression of ERRγ via infection with Ad-*Esrrg* significantly elevated the mRNA levels and extracellular secreted protein levels of MMP3 and MMP13, without affecting

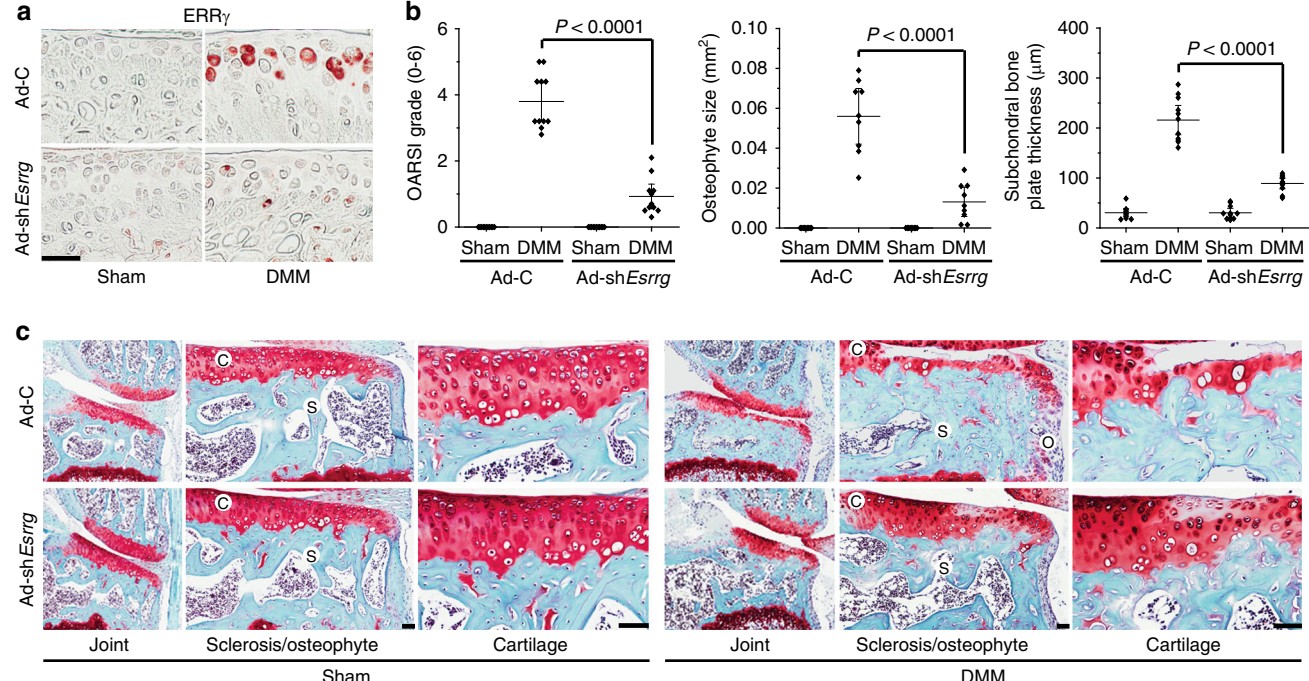

**Fig. 6** Silencing of *Esrrg* in joint tissues abrogates OA pathogenesis in mice. **a** Immunostaining for ERRγ in cartilage sections of DMM-operated mice given an IA injection of Ad-C or Ad-sh*Esrrg*. **b**, **c** WT mice subjected to sham operation or DMM surgery were IA-injected with Ad-C (n = 11 mice) as a control or Ad-sh*Esrrg* (n = 11 mice) to knockdown ERRγ in joint tissues. Cartilage sections were subjected to safranin-O staining (**c**), and determination of OARSI grade, osteophyte size, and subchondral bone plate thickness at 8 weeks after sham operation or DMM surgery (**b**). Values are presented as means ± s.e. m. Two-tailed *t* test and Mann–Whitney *U* test. Scale bar: 50 μm

ADAMTS4 or -5 (Fig. 7a and Supplementary Figs. 7a, b, and 9). In contrast, mRNA levels of SOX9, type II collagen (Coll-II), and aggrecan were reduced by overexpression of ERRγ, although the degree of inhibition was small (Supplementary Fig. 7a, b). Consistent with these findings, MMP3 and MMP13 proteins were upregulated in Ad-*Esrrg*-injected cartilage tissue, whereas SOX9 and type II collagen protein levels were markedly reduced (Fig. 7b). We used chromatin immunoprecipitation (ChIP) assays to examine whether ERRγ directly regulates the expression levels of *Mmp3* and *Mmp13* in articular chondrocytes. Our nucleotide sequence analysis identified a number of putative ERR response elements (ERRE, TCAAGGTCA)[40] in the promoter regions of *Mmp3* and *Mmp13* (Fig. 7c). ChIP assays revealed that ERRγ was bound to ERRE #1 (−2093 to −2083) and ERRE #2 (−2489 to −2478) in the *Mmp3* promoter, and to ERRE #1 (−1094 to −1291) and #2 (−1676 to −1668), but not ERRE #3 (−2389 to −2377), in the *Mmp13* promoter (Fig. 7d; Supplementary Fig. 7c). These findings suggest that *Mmp3* and *Mmp13* are direct target genes of ERRγ in mouse articular chondrocytes.

**Inhibition of ERRγ blocks experimental OA in mice.** Next, we examined whether ERRγ could be a therapeutic target for OA. Because IL-6, an upstream signaling component involved in regulating ERRγ expression, upregulates MMP3 and MMP13 in chondrocytes[33], we first examined whether IL-6-induced upregulation of MMP3 and MMP13 requires the transcriptional activity of ERRγ. We found that treatment with GSK5182, an inverse agonist of ERRγ[22], significantly inhibited IL-6-induced upregulation of MMP3 and MMP13 in primary cultured chondrocytes (Fig. 7e). To validate ERRγ as a potential therapeutic target for OA, we intraperitoneally administered sham- and DMM-operated mice with GSK5182 (20 mg kg−1 body weight). We found that GSK5182 significantly reduced all DMM-induced manifestations of OA, including cartilage destruction, osteophyte

development, and subchondral bone sclerosis (Fig. 7f). These results collectively suggest the possibility that ERRγ could serve as a therapeutic target for OA.

Finally, although mRNA and protein levels of ERRα were unchanged in OA cartilage of human patients and mouse models, we examined whether overexpression of ERRα via IA injection of Ad-*Esrra* in joint tissues causes cartilage damage. IA injection of Ad-*Esrra* caused overexpression of ERRα in cartilage (Supplementary Fig. 8a). However, this overexpression of ERRα did not cause any detectable cartilage damage or synovial inflammation (Supplementary Fig. 8b, c), clearly indicating that ERRγ, but not ERRα, functions as a catabolic regulator of OA in mice.

**Discussion**

The ERR isoforms, ERRα, ERRβ, and ERRγ, regulate various physiological functions, either in common or in an isoform-specific manner. However, whether ERRs regulate OA pathogenesis in vivo has not been investigated to date. Here, we demonstrated that, among ERR isoforms, ERRγ regulates OA pathogenesis in mice by causing cartilage destruction, osteophyte formation, and subchondral bone sclerosis. The results of both gain-of-function (adenovirus-mediated overexpression of ERRγ in mouse joint tissues or *Col2a1-Esrrg* Tg mice) and loss-of-function (*Esrrg*+/− mice or shRNA-mediated silencing of *Esrrg*) experiments clearly indicate that ERRγ functions as a critical catabolic regulator of OA pathogenesis.

The role of ERRs in cartilage biology has not been extensively studied, and most existing studies have focused on ERRα[26]. Among the earliest evidence supporting a possible functional association of ERRα with cartilage development was the spatial and temporal correlation of ERRα expression with cartilage development[41]. Although a relatively large number of studies have reported on the role of ERRα in chondrogenesis and cartilage development[26–28], only a few reported the possible role of

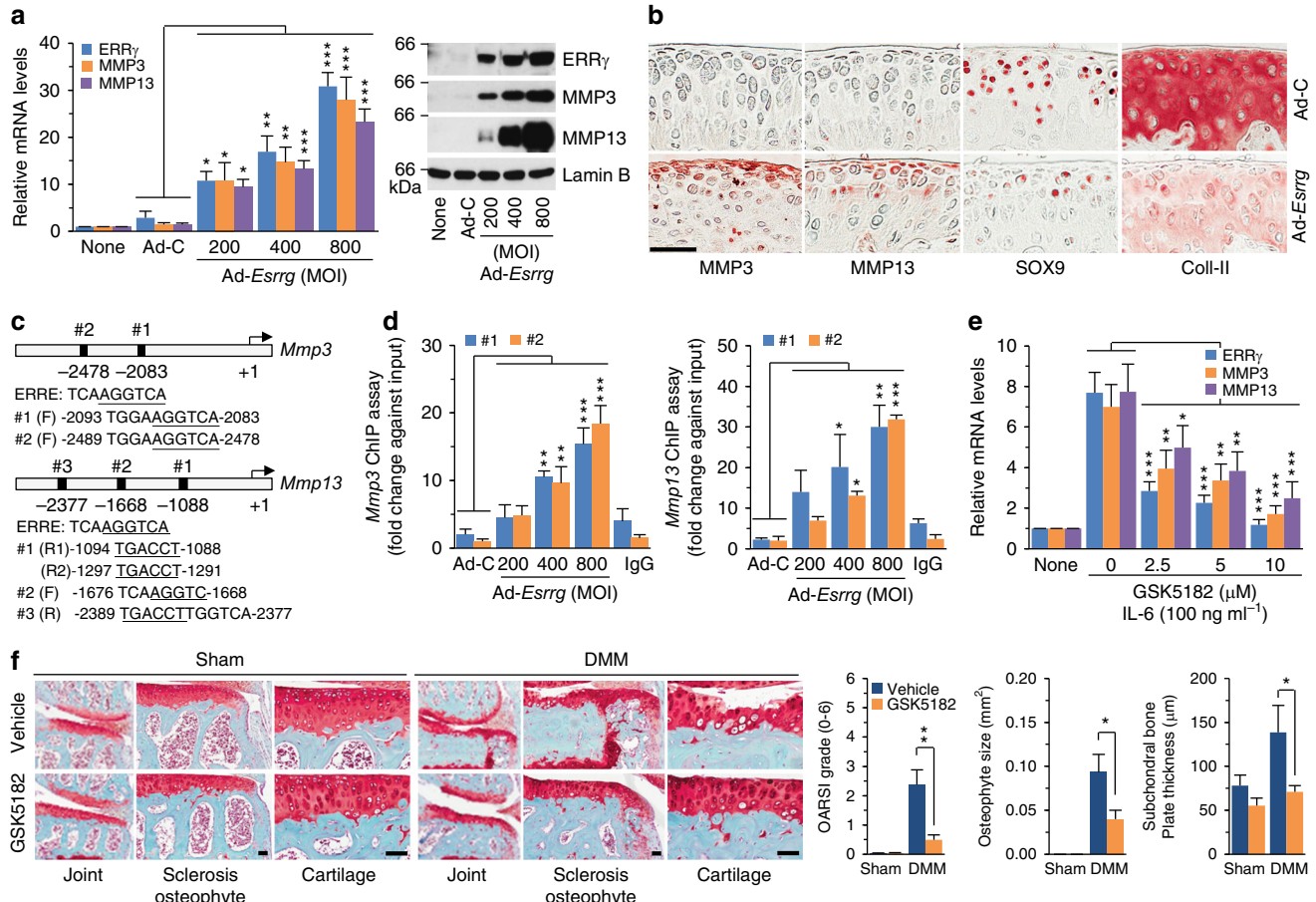

**Fig. 7** ERRγ directly upregulates MMP3 and MMP13 in chondrocytes. **a** The mRNA (left) and protein (right) levels of ERRγ, MMP3, and MMP13 in primary cultured chondrocytes infected with Ad-C (800 MOI) or the indicated MOI of Ad-*Esrrg* for 36 h ($n = 12$). **b** Representative images showing immunostaining for MMP3, MMP13, SOX9, and type II collagen (Coll-II) in cartilage sections of mice obtained after 3 weekly IA injections with Ad-C or Ad-*Esrrg* ($n = 12$ mice per group). **c** The ERRE sequences in the promoter regions of *Mmp3* and *Mmp13* are indicated. **d** Quantitative ChIP assays were performed, and the data are presented as fold-changes relative to each input ($n \geq 4$). **e** qRT-PCR ($n = 11$) analyses of MMP3 and MMP13 expression in chondrocytes treated with IL-6 (100 ng ml$^{-1}$) in the presence of the indicated concentrations of GSK5182, an inverse agonist of ERRγ. **f** Sham-operated or DMM-operated mice were IP-injected with GSK5182 and killed 8 weeks after the operation. Representative images of safranin-O staining and scoring of OARSI grade, osteophyte size, and subchondral bone plate thickness ($n = 10$ mice per group). Values are presented as means ± s.e.m. (*$P < 0.05$, **$P < 0.005$, and ***$P < 0.0005$). One-way ANOVA (**a**,**d**,**e**), two-tailed *t* test (**f**), and Mann–Whitney *U* test (**f**). Scale bar: 50 μm

ERRα in cartilage homeostasis and OA pathogenesis. Bonnelye et al.[32] identified possible in vitro OA-related functions of ERRα, reporting that ERRα expression, which is lower in OA than normal cartilage, is upregulated by IL-1β in OA chondrocytes via signaling pathways involving cyclooxygenase 2, prostaglandin E2, cAMP, and protein kinase A. They also found that ERRα regulates the expression of MMP13 in primary cultured OA chondrocytes in vitro. However, ERRα function in OA pathogenesis in vivo has not been investigated. Furthermore, we found no previous report indicating a role for ERRγ in OA pathogenesis. Our results clearly revealed specific upregulation of ERRγ, among ERR isoforms, in OA cartilage of human patients and DMM-induced mouse OA cartilage. Moreover, in vivo studies using mouse models demonstrated the catabolic functions of ERRγ in OA pathogenesis. Because ERR-α and ERR-γ often display opposite effects in various physiological processes, such as epithelial-to-mesenchymal transition, Warburg effect, and value as a prognosis factor in breast tumors[17,18,25], we also examined possible ERRα functions. However, we found that ERRα expression was not modulated in OA cartilage of humans or mice. Furthermore, overexpression of ERRα in joint tissues via IA injection of Ad-*Esrra* did not cause cartilage damage, indicating

that ERRγ, but not ERRα, functions as a catabolic regulator of OA pathogenesis.

Among the interesting findings in the current study is the observed inhibition of IL-6-induced ERRγ upregulation in chondrocytes by treatment with the ERRγ inverse agonist GSK5182. This suggests possible autoregulation of ERRγ in chondrocytes. Indeed, we have previously demonstrated that ERRγ protein levels are decreased by GSK5182 treatment[42], which might reflect the operation of an autoregulatory feed-forward mechanism in the regulation of ERRγ gene expression[43]. Similar to our observation, it has been shown that IL-1β-induced upregulation of ERRα in chondrocytes is reduced by the ERRα inverse agonist XCT790[32]. This suggests that both ERRα and ERRγ expression in chondrocytes are regulated by positive feedback mechanisms. We have previously reported amplification of HIF-2α catabolic signaling in chondrocytes during OA pathogenesis. For instance, HIF-2α signaling leading to the production of the matrix-degrading enzymes, MMP3 and MMP13[10], is amplified by the HIF-2α targets, IL-6[32] and NAMPT (visfatin)[11]. Additionally, the NAMPT downstream target SIRT also stimulates HIF-2α transcriptional activity by stabilizing HIF-2α protein[44]. Furthermore, two essential catabolic

signaling pathways in OA pathogenesis—HIF-2α and the zinc-ZIP8-MTF1 axis—reciprocally activate each other, resulting in amplification of catabolic signaling[45]. Collectively, these observations suggest that catabolic signaling processes involved in OA pathogenesis are amplified upon onset of the disease.

The only study to date describing a role for ERRγ in cartilage is one by Cardelli et al.[30], who reported that cartilage-specific overexpression of ERRγ results in chondrodysplasia and reduced chondrocyte proliferation. These authors found a reduction in the length of the femur and tibia in Tg mice, with no differences in intramembranous ossification. They additionally found that the overexpression of ERRγ impaired chondrocyte proliferation, leading to a decrease in the size of the proliferative zone in the growth plate[30]. However, our Tg mice exhibited no marked differences in the whole skeleton of embryos (E18.5) and length of resting/proliferative and hypertrophic zones in 2-week-old mice. Although it is not clear what accounts for these differences, they may be attributable to differences in expression systems[46] and expression levels of target genes. We previously reported that cartilage-specific Dickkopf-1 Tg mice, generated using the same system as the current Tg mice, exhibit reduced experimental OA[47] without differences in cartilage or bone development[46]. We also showed that cartilage-specific Tg mice generated by our system, including those overexpressing HIF-2α[10], NAMPT[11], BATF[35], or ZIP8[12], exhibited enhanced experimental OA without detectable differences in skeletal development. In addition to the difference in expression system, we also found that the expression level of ERRγ protein in our Tg mice was less than that in mice generated by Cardelli et al.[30]. We postulate that these differences in the expression levels of ERRγ contribute to the observed differences. Nevertheless, the enhanced expression of matrix-degrading enzymes in cartilage of Tg mice compared with WT mice appears to be responsible for the enhanced DMM-induced OA pathogenesis in Tg mice.

The prevalence of OA is higher among women than men, and this prevalence increases after menopause. Various studies using animal models indicate that estrogen and estrogen signaling are associated with OA pathogenesis[48]. For instance, mice lacking estrogen receptor-α develop more and larger osteophytes and a thinner lateral subchondral plate[49]. A lack of estrogens also increases subchondral bone remodeling[50]. However, although the term "estrogen-related receptor (ERR)" reflects the molecular origin of the three family members, ERRs are functionally different from estrogen receptors. For instance, ERRs do not bind to natural estrogens, nor are they directly involved in classic estrogen-signaling pathways or biological processes[18,24]. Furthermore, ERRs and estrogen receptors display strict binding site specificities[51], with ERRs exhibiting a marked preference for the sequence TCAAGGTCA, an ERR response element (ERRE). Furthermore, unlike estrogen receptors, ERRs do not require ligand binding for their transcriptional activity[18,24]. Therefore, the catabolic functions of ERRγ in OA pathogenesis might not be related to estrogen, even though estrogen is known to be associated with OA pathogenesis[48].

Cartilage destruction is one of the most prominent characteristics of OA and is primarily caused by the upregulation of matrix-degrading enzymes, such as MMP3, MMP13, and ADAMTS5[5–8]. These matrix-degrading enzymes originate from chondrocytes of cartilage tissue or from synovial cells, such as fibroblast-like synoviocytes (FLS)[5]. Our approaches using *Esrrg*[+/−] mice, shRNA-mediated silencing, and adenovirus-mediated overexpression affected ERRγ expression in all joint tissues, limiting our ability to identify specific joint tissues associated with OA pathogenesis. We observed, however, that ERRγ was upregulated in the cartilage tissue, but not the synovial tissue, of DMM-operated mice, and that ERRγ caused upregulation of

MMP3 and MMP13 in primary cultured articular chondrocytes. These results strongly support the interpretation that ERRγ causes OA pathogenesis by upregulating MMP3 and MMP13 in articular chondrocytes. We also found that ERRγ downregulated SOX9, type II collagen, and aggrecan, although the degree of downregulation was small. Interestingly, compared with in vitro effects of ERRγ overexpression, ERRγ overexpression via IA injection of Ad-*Esrrg* in cartilage tissue caused marked decreases in the protein expression levels of SOX9 and type II collagen. The decrease in type II collagen might reflect its MMP-mediated degradation in ERRγ-overexpressing cartilage tissue. Thus, it may be possible that the reduction in SOX9 expression in Ad-*Esrrg*-injected cartilage is attributable to both direct and indirect actions of ERRγ. In addition to cartilage destruction, OA manifestations include osteophyte formation and subchondral bone sclerosis, reflecting the fact that OA is a whole-joint disease affecting all tissues of the joint, including cartilage, synovial tissues, and subchondral bone[2]. However, the relative contributions, relationships, and key tissues involved in each event that occurs during OA pathogenesis have not been elucidated[2,3]. For instance, it is not yet clear whether subchondral bone sclerosis is a driving force for cartilage breakdown or a consequence of articular cartilage catabolism. Therefore, the regulatory mechanisms of ERRγ in osteophyte formation and subchondral bone sclerosis remain to be elucidated.

Here, we demonstrated that intraperitoneal administration of GSK5182 reduces DMM-induced OA pathogenesis, although it remains to be clarified whether this is attributable to inhibition of ERRγ in the tissues, since intraperitoneal administration might affect ERRγ in the whole body. However, although cartilage is an avascular tissue, OA is not merely a local disease confined to the joint, and many soluble mediators that regulate OA pathogenesis are found in circulating blood[52]. Additionally, the human pathology may differ in certain respects from that in mouse models. However, we found that expression patterns of ERRγ as well as ERRα and ERRβ were similar in OA cartilage of human patients and various mouse models. Our results are the first to demonstrate that ERRγ acts as a catabolic regulator of cartilage degeneration and OA pathogenesis, and collectively support the idea that ERRγ could be a therapeutic target for OA.

## Methods

**Human OA cartilage and experimental OA in mice**. Human OA cartilage was sourced from individuals undergoing arthroplasty[10–13]. The Institutional Review Board of Wonkwang University Hospital approved the use of these materials, and all participants provided written informed consent before the operative procedure. C57BL/6 J mice were used for the experimental OA studies. *Esrrg*[+/−] mice (B6.129P2-*Esrrg*[tm1Dgen]/Mmnc) were purchased from the Mutant Mouse Regional Resource Center (MMRRC). *Col2a1-Esrrg* Tg mice were generated using the *Col2a1* promoter and enhancer, as previously described[10–12,45]. All experiments were approved by the Gwangju Institute of Science and Technology Animal Care and Use Committee. Experimental OA was induced in 10-week-old male mice by DMM surgery, or by IA injection (once weekly for 3 weeks) of adenovirus ($1 \times 10^9$ plaque-forming units [PFUs] in a total volume of 10 μl) expressing ERRγ (Ad-*Esrrg*), ERRα (Ad-*Esrra*), ZIP8 (Ad-*Slc39a8*), or HIF-2α (Ad-*Epas1*)[10–13,29]. Where indicated, male mice of various ages (8, 12, 16, and 20 weeks old) were used for experimental OA studies. Mice were killed at 8 weeks after DMM surgery or at 3 or 8 weeks after the first IA injection, and subjected to histological and biochemical analyses. Intraperitoneal injection of GSK5182 (20 mg kg[−1] body weight) was performed every 2 days for 7 weeks, beginning 1 week after DMM surgery.

**Histology and immunohistochemistry**. Human OA cartilage was frozen, sectioned at a thickness of 5 μm, and fixed in paraformaldehyde. Sulfate proteoglycans were detected by alcian blue staining, and cartilage destruction in mice was examined using safranin-O staining. Briefly, knee joints were fixed in 4% paraformaldehyde, decalcified in 0.5 M EDTA, and embedded in paraffin. The paraffin blocks were sectioned at a thickness of 5 μm, and sections were deparaffinized in xylene, hydrated with graded ethanol, and stained with safranin-O. Cartilage destruction was scored by five observers under blinded conditions using the OARSI scoring system (grade 0–6)[10–13,53]. The results of OARSI grade scoring represent the mean of the maximum score in each mouse, and the representative safranin-O-

stained image was selected from the most advanced lesion among serial sections. Synovitis was determined by safranin-O and hematoxylin staining, and synovial inflammation was scored (grade 0–3) as described previously[10–13]. Osteophyte formation was identified by safranin-O staining, and osteophyte size was measured with an Aperio Image Scope V12 (Leica Biosystems)[12,13]. Subchondral bone sclerosis was determined by measuring the thickness of the subchondral bone plate[12,13]. ERRγ was immunostained using 1 μg ml$^{-1}$ of mouse monoclonal antibody (PP-H6812-00; R&D Systems Inc.) or 0.5 μg ml$^{-1}$ of rabbit polyclonal antibody (ab49129; Abcam). Additional immunostaining was performed using antibodies against ERRα (40 μg ml$^{-1}$; sc-65715; Santa Cruz Biotech), MMP3 (4 μg ml$^{-1}$; ab52915; Abcam), MMP13 (1:100 dilution; ab51072; Abcam), SOX9 (12 μg ml$^{-1}$; ab185966; Abcam), and type II collagen (2 μg ml$^{-1}$; MAB8887; Millipore).

**Primary culture of articular chondrocytes.** Femoral condyles and tibial plateaus were obtained from mice, and chondrocytes were isolated from cartilage tissue by digestion with 0.2% collagenase[10–13,54]. The cells were maintained as a monolayer in Dulbecco's modified Eagle's medium (DMEM; Gibco) supplemented with 10% fetal bovine serum and antibiotics (penicillin G and streptomycin). On culture day 2, cells were infected with 800 MOI (multiplicity of infection) of empty adenovirus (Ad-C); infected with Ad-Esrrg, Ad-Epas1, or Ad-Slc39a8 at various MOIs; or treated with the inflammatory cytokines, IL-1β (1 ng ml$^{-1}$) or IL-6 (100 ng ml$^{-1}$) for the indicated time periods, unless otherwise indicated in each experiment.

**Infection and IA injection of adenovirus.** The adenoviruses expressing mouse ERRγ (Ad-Esrrg), Ad-shEsrrg, and ERRα (Ad-Esrra) were as previously described[22]. The adenoviruses expressing mouse HIF-2α (Ad-Epas1) and ZIP8 (Ad-Slc39a8) were purchased from Vector Biolabs[10,12]. Mouse articular chondrocytes were cultured for 2 days, infected with the indicated MOI of adenovirus for 2 h, and cultured in the presence or absence of pharmacological agents for an additional 36 h. For IA injection of adenovirus, Ad-Esrrg, Ad-shEsrrg, Ad-Esrra, or Ad-C (1 × 10$^9$ PFUs in a total volume of 10 μl) was injected into the knee joints of mice once per week for 3 weeks.

**Skeletal staining and histological analysis.** Skeletons of whole-mouse embryos were stained with alcian blue and alizarin red, as described previously[10–13,46]. Briefly, whole embryos were skinned, eviscerated, fixed with 95% ethanol for 4 days, and immersed in acetone for 3 days. Samples were stained with one volume of 0.3% alcian blue 8GX in 70% ethanol, one volume of 0.1% alizarin red S in 95% ethanol, one volume of 100% acetic acid, and 17 volumes of 100% ethanol. Bone histology was analyzed as described previously[46]. Briefly, metatarsals of 2-week-old mice were fixed in 4% paraformaldehyde, decalcified in 0.5 M EDTA (pH 7.4) for 7 days at 4 °C, and embedded in paraffin. The deparaffinized sections were stained with 1% alcian blue in 0.1 N HCl for 10 minutes, and washed with 0.1 N HCl for 5 minutes. The lengths of the resting/proliferative and hypertrophic zones were measured using a microscope image-analysis program (AxioVision).

**Conventional RT-PCR and qRT-PCR.** Total RNA was extracted from primary cultured chondrocytes using the TRI reagent (Molecular Research Center, Inc.). For cartilage tissues, mouse knee joints were scraped with a blade after DMM surgery or sham operation, and RNA was isolated using the TRI reagent[10–13]. Total RNA was reverse transcribed, and the resulting cDNA was PCR-amplified using the PCR primers and experimental conditions summarized in Supplementary Table 2. qRT-PCR was performed using an iCycler thermal cycler (Bio-Rad) and SYBR premixExTaq reagents (TaKaRa Bio). For each target gene, transcript levels were normalized to that of glyceraldehyde-3-phosphate dehydrogenase (GAPDH) and expressed as fold-change relative to the indicated control.

**Western blotting.** Total cell lysates were prepared in lysis buffer (150 mM NaCl, 1% NP-40, 50 mM Tris, 0.2% sodium dodecyl sulfate [SDS], and 5 mM NaF) and used to detect ERRγ. Secreted proteins (MMP3 and MMP13) were detected after trichloroacetic acid (TCA) precipitation of proteins from 900 μl of serum-free conditioned medium. All lysis buffers contained a cocktail of protease inhibitors and phosphatase inhibitors (Roche). The following antibodies were used for Western blotting: rabbit polyclonal anti-ERRγ (sc-66883) from Santa Cruz, and mouse monoclonal anti-MMP3 (clone EP1186Y) and anti-MMP13 (clone EP1263Y) from Epitomics.

**ChIP assay.** ChIP assays were performed using a Pierce Agarose ChIP Kit (Thermo Scientific)[35], as described by the manufacturer. Briefly, 70% confluent mouse articular chondrocytes were infected with Ad-Esrrg at an MOI of 800 for 36 h. The primers for ChIP assays were designed to amplify two different ERRE-containing regions of the Mmp3 promoter and three regions of the Mmp13 promoter (Supplementary Table 3). DNA and proteins were cross-linked by incubating cells with 1% formaldehyde for 10 minutes at room temperature. Excess formaldehyde was quenched by incubating with glycine for 5 minutes. Cells were lysed and nuclei were digested using micrococcal nuclease. Sheared chromatin was diluted and immunoprecipitated with 2 μg of anti-ERRγ or control IgG antibody. DNA–protein complexes were eluted from protein A/G agarose beads using a spin column, and then reverse cross-linked by incubation with NaCl at 65 °C. The relative binding of ERRγ to the ERRE regions of Mmp3 and Mmp13 promoters was analyzed by PCR amplification using a Mastercycler thermal cycler. For quantitative ChIP assays, qRT-PCR was performed using an iCycler thermal cycler (Bio-Rad) and SYBR premixExTaq reagents (TaKaRa Bio). The data represent fold-changes relative to each input.

**Microarray analyses.** Total RNA was extracted from mouse articular chondrocytes using a Purelink RNA mini kit (Ambion). The concentration, purity, and integrity of the extracted RNA were determined by spectrophotometry. Four replicates for each cell type were isolated and processed. RNA from mouse chondrocytes was analyzed using Affymetrix Gene Chip arrays (Affymetrix GeneChip Mouse Gene 2.0 ST Array) in accordance with the Affymetrix protocol (Macrogen Inc., Seoul, Korea). Probe signals in the raw data were normalized by the RMA (robust multiarray average) for each separate data set (IL-1β treatment or injection of Ad-C, Ad-Epas1, or Ad-Slc39a8). Normalization was performed using R v.3.3.2 with Affy package v.1.52.0. To find DEGs (differentially expressed genes), we performed the Student's t-test followed by the Benjamini–Hochberg multiple hypothesis test for each group using python v.3.4.3 and StatsModels library v.0.8.0. The cutoff values for DEG identification were adjusted such that the $P$-value was less than 0.05 (FDR < 0.05) and the absolute value of log fold change was greater than 1 (|LogFC| > 1).

**Statistical analysis.** All statistical analyses were performed using IBM SPSS Statistics 21 software. For cell-based in vitro studies, two-tailed Student's t-tests with unequal sample sizes and variances, and two-way analysis of variance (ANOVA) with post hoc tests (LSD) were used for pairwise comparisons and multi-comparisons, respectively. Data collected from mouse experiments were analyzed using the nonparametric Mann–Whitney U test. The n-value indicates the number of independent experiments or mice. Significance was accepted at the 0.05 level of probability ($P < 0.05$).

**Data availability.** Microarray data have been deposited in Gene Expression Omnibus with the accession codes GSE104794 (for HIF-2α), GSE104795 (for ZIP8), and GSE104793 (for IL-1β). All other data supporting the findings of this study are available within the paper and its supplementary information files.

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

## Acknowledgements

This work was supported by grants from the National Research Foundation of Korea (2016R1A3B1906090 and 2016R1A5A1007318), the Korea Health Technology R&D Project through the Korea Health Industry Development Institute (HI16C0287 and HI14C3484), and the GIST Research Institute (GRI) in 2017. Y.-O.S. was supported by a grant from the Basic Science Research Program through the National Research Foundation of Korea (2016R1D1A1B03930327). H.-S.C. was supported by a National Creative Research Initiatives Grant (20110018305) through the National Research Foundation of Korea.

## Author contributions

Y.-O.S.: study design, data acquisition, data analysis, data interpretation, manuscript preparation, and manuscript approval. S.-K.P., J.-S.K., Y.W., W.-S.C., J.R., J.-H.R., D.-K.K. and H.-S.C.: data acquisition, data analysis, and manuscript approval. C.-H.C. provided and evaluated the human joint samples. J.-S.C. (jschun@gist.ac.kr) takes responsibility for the integrity of this work.

## Additional information

**Competing interests:** The authors declare no competing financial interests.

