## [Peer Review File · Nature Communications]

Reviewers' comments:

Reviewer #1 (Remarks to the Author):

This paper is high interesting and important to the OA field, but warrant additional clarifications and further validation.

For references to human cartilage among other papers this review is cited " . Constitutive activities of estrogen-related receptors: transcriptional regulation of metabolism by the ERR pathways in health and disease" – which is a review not containing any patient material. For human studies, careful description of patients and materials must be included.

The use of very young mice in OA research is associated with many developmental effects. All the experiments done, would need to be reproduced in older animals, to support the impressive data seems in young animals.

A substantial amount of estrogen receptor work has been undertaken in the bone field, many of these papers (in particular by Dr. Sims and Dr. Baron) should be discussed and cited.

Reviewer #2 (Remarks to the Author):

In their manuscript Son et al. describe the induction of ERRgamma expression in osteoarthritis (OA) and document the effect of this nuclear receptor on parameters of OA. Several mouse models of OA are used as well as several models of ERRgamma modulation (transgenic and +/- animals, in vivo adenovirus-mediated overexpression or inhibition), rendering the conclusions convincing. Few things are known about OA and the authors' demonstration will be helpful in our understanding of the mechanisms of this disease as well as to propose ERRgamma as a potential target to be deactivated in OA.

The results of the experiments are clear and the paper is nicely written (although the Discussion section may deserved to be extended).

Complementary experiments suggested below may reinforce the validity of the functional links proposed by the authors.

Major points:

1 - The relationship between human pathology and mouse models in the present context is unclear. Indeed not only ERRg but also ERRa is stimulated in the human, but not in mouse. ERRa and g often display opposite effect (e.g. in EMT, Warburg effect, as prognosis factor in breast tumors); it may be that their effects on OA compensate each other's in human, but not mouse, questioning the validity of mouse models for studying ERRs' effect in OA, a point that should be discussed. The authors should show the basal relative expressions of ERRa and g (at the mRNA and protein levels), in damaged and undamaged human OA cartilage (i.e. it may be that ERRa expression is negligible relative to ERRg, ruining my above concern). Also: is the same discrepancy observed when comparing human to mouse cartilage cells in vitro (upon IL6 exposure for instance)? Whatever the answer to these questions, the relevance of the present data to human pathology should be more discussed.

2 - There are missing links in the authors' demonstration of an IL1b/IL6/ERRg axis (Fig 2).

Authors could show that:

- IL6 protein is indeed secreted upon IL1b exposure in their system
- Blocking IL6 impairs ERRg increased expression upon IL1b
- IL1b treatment increases ERRg expression in their system. This data should be available in their microarray study, but could be confirmed by qPCR; by the way what is the status of ERRa and b expression (and potential variations) in their microarray.
- Is the effect of IL6 on ERRg also JAK2/STAT3-dependent as in the liver? Authors could use the AG490 JAK2 inhibitor to show this.

- The time point of maximal ERRg induction by IL6 seems discrepant between the two panels of Fig 2b. Any comment?
- GSK5182 reduces IL6-dependent ERRg increased expression (Fig 7f). Is ERRg autoregulated at the transcriptional level?

Besides, IL1b elicits a much stronger increase in IL6 expression than do HIF2 or ZIP8 overexpression, at least at the microarray.

- Is this confirmed by independent qPCR and at the level of secreted IL6 protein?
- If yes, would the lower amount of IL6 produced upon HIF2 or ZIP8 overexpression be sufficient to induce ERRg (as well as MMP3/MMP13)?

3 - There are missing links in the relationships with MMP3/MMP13.

- Effect of IL6 on MMP3/13 should be shown at the protein level (Fig 2)
- In their experiments (fig 7b), do the authors examine secreted/processed MMP3/MMP13 proteins (that is, examining culture medium instead of cellular extracts)?
- What is the status of MMP3/13 in ERRgTg and ERRg+/- animals (when induced by IL1b or IL6 for cultured chondrocytes or in DMM challenged animals)

Minor points:

- The reviews (ref 15, 24, 25) are not appropriate to reference the fact that IL6 stimulates ERRg expression. The Kim paper (ref 30) is more appropriate.
- A comment is lacking at the end of first chapter in Results section to explain why the authors examine MMP3 and MMP13 in Fig 2b. Besides this result is not commented per se.
- The difference between wt and Tg animals after DMM (fig 4d) could be better illustrated: show what we have to look at with arrows for example.
- Fig 5 should not be labeled with "KO" since heterozygote animals are used (+/- would be more appropriate)
- Cardelli et al (2013) reported no effect on Sox9 and increased Col2 expression in ERRgTg animals. This discrepancy could be discussed.
- First sentence in Discussion section is too straightforward. i- authors do not distinguish between the different ERR; ii- muscular effects of ERRs were only shown in cell culture systems (as far as I know); ii- ERR effects in bone were shown in mouse models and are unclear in the human
- Discussion section is really short and could deserve extension.

JM Vanacker, reviewer

Reviewer #3 (Remarks to the Author):

In this manuscript, the authors examined the role of the orphan nuclear receptors estrogen-related receptors (ERRs) in osteoarthritis (OA) pathogenesis. Using both gain-of-function and loss-of-function approaches, the authors provide ample evidence to suggest ERRγ contributes to OA pathogenesis, in part through catabolic regulation of MMP13 and MMP3 expression. As the roles of ERRs in OA remain previously unexplored, this finding gives good insight into the understanding of nuclear receptor function in OA and suggests ERRγ to be a novel therapeutic target. However, certain fundamental concerns exist regarding the presentation and discussion of the results that should be addressed.

1. In the introduction, the authors refer to findings from reference #22 where cartilage-specific over-expression of ERRγ has previously been shown to result in chondrodysplasia and reduced chondrocyte proliferation. This contradicts the results in Figure 3 showing no defects in skeletal development in the cartilage-specific over-expressors generated here. Further studies should be conducted at multiple time-points (instead of just at E18.5) throughout development to confirm ERRγ overexpression to have no effect on skeletal development and chondrocyte differentiation, as

these effects could potentially alter the conclusions made from these animals in the context of OA.

2. The authors should consider showing more representative images of ERR γ IHC from Ad-C and Ad-Esrrg sections, as the degree of adenovirus penetration into the articular cartilage is not clear from some images (e.g. Fig. 3a only shows the meniscus and very superficial layers of the cartilage and not deeper layers of cartilage and the subchondral bone).

3. It is not clear whether the OARSI scoring used was modified in any way, as certain scores seem higher than expected compared to their corresponding histological images (e.g. Fig 5f, 5g show only proteoglycan loss and no actual structural damage which would result in a much lower score). Please indicate any scoring modifications in the methods section or provide more representative histological images to avoid confusion. More generally some of the OA phenotypes mentioned appear rather mild, consisting largely of proteoglycan depletion without damage to the cartilage surface.

4. Please provide more representative histological images that show the whole joint in OA studies. Often the images are very cropped and only show one area of the cartilage, or only show the femur and not both the femur and tibia (e.g. Fig 4d vs Fig 5d). Please be consistent in presenting histological data to avoid bias.

5. It is not clear why 8-week old animals were chosen for OA induction through DMM surgery or IA injections. Please discuss thoroughly the rationale behind using animals this young and why only one time-point post-surgery/post-injection was examined. 8 week-old mice are not mature skeletally and therefore suboptimal for these studies.

6. The authors should provide an explanation for why the inverse agonist (GSK5182), instead of shRNA, was used in vitro (Fig. 7f), and why this inverse agonist caused decreased mRNA levels of ERR γ . The authors should consider conducting further studies to examine potential feedback regulation of ERR γ expression, as well as validating the effect of GSK5182 in vivo.

7. Please perform quantitative real-time PCR and quantitative ChIP analysis. Changes in mRNA levels of Adamts4, Adamts5, Sox9 and Col2a1 may not be detected through qualitative PCR, and could be the reason behind the observed decrease in protein levels of Sox9 and Col2a1.

Reviewers' comments:

Reviewer #1 (Remarks to the Author):

This paper is high interesting and important to the OA field, but warrant additional clarifications and further validation. For references to human cartilage among other papers this review is cited. "Constitutive activities of estrogen-related receptors: transcriptional regulation of metabolism by the ERR pathways in health and disease" – which is a review not containing any patient material. For human studies, careful description of patients and materials must be included. The use of very young mice in OA research is associated with many developmental effects. All the experiments done, would need to be reproduced in older animals, to support the impressive data seems in young animals. A substantial amount of estrogen receptor work has been undertaken in the bone field, many of these papers (in particular by Dr. Sims and Dr. Baron) should be discussed and cited.

Reviewer #2 (Remarks to the Author):

In their manuscript Son et al. describe the induction of ERR γ expression in osteoarthritis (OA) and document the effect of this nuclear receptor on parameters of OA. Several mouse models of OA are used as well as several models of ERR γ modulation (transgenic and +/- animals, in vivo adenovirus-mediated overexpression or inhibition), rendering the conclusions convincing. Few things are known about OA and the authors' demonstration will be helpful in our understanding of the mechanisms of this disease as well as to propose ERR γ as a potential target to be deactivated in OA. The results of the experiments are clear and the paper is nicely written (although the Discussion section may deserved to be extended). Complementary experiments suggested below may reinforce the validity of the functional links proposed by the authors.

Major points:

1. The relationship between human pathology and mouse models in the present context is unclear. Indeed not only ERR γ but also ERR α is stimulated in the human, but not in mouse. ERR α and γ often display opposite effect (e.g. in EMT, Warburg effect, as prognosis factor in breast tumors); it may be that their effects on OA compensate each other's in human, but not mouse, questioning the validity of mouse models for studying ERRs' effect in OA, a point that should be discussed. The authors should show the basal relative expressions of ERR α and γ (at the mRNA and protein levels), in damaged and undamaged human OA cartilage (i.e. it may be that ERR α expression is negligible relative to ERR γ , ruining my above concern). Also: is the same discrepancy observed when comparing human to mouse cartilage cells in vitro (upon IL-6 exposure for instance)? Whatever the answer to these questions, the relevance of the present data to human pathology should be more discussed.
2. There are missing links in the authors' demonstration of an IL-1 β /IL-6/ERR γ axis (Fig. 2). Authors could show that:
 - IL6 protein is indeed secreted upon IL-1 β exposure in their system
 - Blocking IL-6 impairs ERR γ increased expression upon IL-1 β
 - IL-1 β treatment increases ERR γ expression in their system. This data should be available in their microarray study, but could be confirmed by qPCR; by the way

what is the status of ERR α and β expression (and potential variations) in their microarray.

- Is the effect of IL-6 on ERR γ also JAK2/STAT3-dependent as in the liver? Authors could use the AG490 JAK2 inhibitor to show this.
 - The time point of maximal ERR γ induction by IL-6 seems discrepant between the two panels of Fig. 2b. Any comment?
 - GSK5182 reduces IL-6-dependent ERR γ increased expression (Fig. 7f). Is ERR γ autoregulated at the transcriptional level? Besides, IL-1 β elicits a much stronger increase in IL-6 expression than do HIF-2 or ZIP8 overexpression, at least at the microarray.
 - Is this confirmed by independent qPCR and at the level of secreted IL-6 protein?
 - If yes, would the lower amount of IL-6 produced upon HIF-2 or ZIP8 overexpression be sufficient to induce ERR γ (as well as MMP3/MMP13)?
3. There are missing links in the relationships with MMP3/MMP13.
- Effect of IL-6 on MMP3/13 should be shown at the protein level (Fig. 2)
 - In their experiments (Fig. 7b), do the authors examine secreted/processed MMP3/MMP13 proteins (that is, examining culture medium instead of cellular extracts)?
 - What is the status of MMP3/13 in ERR γ Tg and ERR γ +/- animals (when induced by IL-1 β or IL-6 for cultured chondrocytes or in DMM challenged animals)

Minor points:

- The reviews (ref 15, 24, 25) are not appropriate to reference the fact that IL-6 stimulates ERR γ expression. The Kim paper (ref 30) is more appropriate.
- A comment is lacking at the end of first chapter in Results section to explain why the authors examine MMP3 and MMP13 in Fig 2b. Besides this result is not commented per se.
- The difference between Wt and Tg animals after DMM (Fig. 4d) could be better illustrated: show what we have to look at with arrows for example.
- Fig 5 should not be labeled with "KO" since heterozygote animals are used (+/- would be more appropriate)
- Cardelli et al (2013) reported no effect on Sox9 and increased Col2 expression in ERR γ Tg animals. This discrepancy could be discussed.
- First sentence in Discussion section is too straightforward. i- authors do not distinguish between the different ERR; ii- muscular effects of ERRs were only shown in cell culture systems (as far as I know); ii- ERR effects in bone were shown in mouse models and are unclear in the human
- Discussion section is really short and could deserve extension.

Reviewer #3 (Remarks to the Author):

In this manuscript, the authors examined the role of the orphan nuclear receptors estrogen-related receptors (ERRs) in osteoarthritis (OA) pathogenesis. Using both gain-of-function and loss-of-function approaches, the authors provide ample evidence to suggest ERR γ contributes to OA pathogenesis, in part through catabolic regulation of MMP13 and MMP3 expression. As the roles of ERRs in OA remain previously unexplored, this finding gives good insight into the understanding of nuclear receptor function in OA and suggests ERR γ to be a novel therapeutic target. However, certain fundamental concerns exist regarding the presentation and discussion of the results that should be addressed.

1. In the introduction, the authors refer to findings from reference #22 where cartilage-specific over-expression of ERR γ has previously been shown to result in chondrodysplasia and reduced chondrocyte proliferation. This contradicts the results in Figure 3 showing no defects in skeletal development in the cartilage-specific over-expressors generated here. Further studies should be conducted at multiple time-points (instead of just at E18.5) throughout development to confirm ERR γ overexpression to have no effect on skeletal development and chondrocyte differentiation, as these effects could potentially alter the conclusions made from these animals in the context of OA.
2. The authors should consider showing more representative images of ERR γ IHC from Ad-C and Ad-Esrrg sections, as the degree of adenovirus penetration into the articular cartilage is not clear from some images (e.g. Fig. 3a only shows the meniscus and very superficial layers of the cartilage and not deeper layers of cartilage and the subchondral bone).
3. It is not clear whether the OARSI scoring used was modified in any way, as certain scores seem higher than expected compared to their corresponding histological images (e.g. Fig. 5f, 5g show only proteoglycan loss and no actual structural damage which would result in a much lower score). Please indicate any scoring modifications in the methods section or provide more representative histological images to avoid confusion. More generally some of the OA phenotypes mentioned appear rather mild, consisting largely of proteoglycan depletion without damage to the cartilage surface.
4. Please provide more representative histological images that show the whole joint in OA studies. Often the images are very cropped and only show one area of the cartilage, or only show the femur and not both the femur and tibia (e.g. Fig. 4d vs Fig. 5d). Please be consistent in presenting histological data to avoid bias.
5. It is not clear why 8-week old animals were chosen for OA induction through DMM surgery or IA injections. Please discuss thoroughly the rationale behind using animals this young and why only one time-point post-surgery/post-injection was examined. 8 week-old mice are not mature skeletally and therefore suboptimal for these studies.
6. The authors should provide an explanation for why the inverse agonist (GSK5182), instead of shRNA, was used in vitro (Fig. 7f), and why this inverse agonist caused decreased mRNA levels of ERR γ . The authors should consider conducting further studies to examine potential feedback regulation of ERR γ expression, as well as validating the effect of GSK5182 in vivo.
7. Please perform quantitative real-time PCR and quantitative ChIP analysis. Changes in mRNA levels of Adamts4, Adamts5, Sox9 and Col2a1 may not be detected through qualitative PCR, and could be the reason behind the observed decrease in protein levels of Sox9 and Col2a1

Point-to-point response to reviewers' comments

Summary of changes we have made to individual figures.

We originally submitted our manuscript with 7 main figures and 2 supplementary tables. Because of additional experiments performed during the revision process, the revised manuscript contains 7 main figures, 7 supplementary figures, and 3 supplementary tables. We also modified the appropriate text and re-organized the sequence of experimental data in both main and supplemental figures. Changes made to individual figures are summarized below.

Figure 1. Original Fig. 1 with modifications to address reviewer #2's comment 1.

Figure 2. Original Fig. 2 but extensively revised with additional data.

a: Additional data to address reviewer #2's comments 2-1 and 2-7.

b,c: Additional data to address reviewer #2's comments 2-1 and 3-1.

d,e: Original Fig. 2B with additional data to address reviewer #2's comment 2-5 and 3-1.

f: Additional data to address reviewer #2's comment 2-2, 2-3, and 2-4.

Figure 3. Original Fig. 3 with modifications to address reviewer #3's comments 2, 3, 4 and 5.

Figure 4. Original Fig. 4 with modifications to address reviewer #3's comments 2, 3, 4 and 5.

Figure 5. Original Fig. 5 with modifications to address reviewer #3's comments 2, 3, 4 and 5.

Figure 6. Original Fig. 6 with modifications to address reviewer #3's comments 2, 3 and 4.

Figure 7. Original Fig. 7 with modifications to address reviewer #3's comments 3, 4, 5 and 6.

Figure S1. Additional data to address reviewer #2's comment 1.

Figure S2. Additional data and re-arrangement of original data to address reviewer #1's comment 2 and reviewer #3's comments 2 and 5.

Figure S3. Additional data and re-arrangement of original data to address reviewer #2's comment 3-3, reviewer #2's minor comment, reviewer #3's comment 1, and reviewer #3's comments 2 and 5.

Figure S4. Additional data to address reviewer #1's comment 2 and reviewer #3's comments 4 and 5.

Figure S5. Additional data and re-arrangement of original data to address reviewer #2's comment 3-3 and reviewer #3's comments 2 and 5.

Figure S6. Additional data and re-arrangement of original data to address reviewer #3's comment 7.

Figure S7. Additional data and re-arrangement of original data to address reviewer #2's comment 1.

Table S1. Original Fig. 2A was converted to Supplementary Table 1 with modifications to address reviewer #2's comment 2-3.

Table S2. Original Supplementary Table 1.

Table S3. Original Supplementary Table 2.

REVIEWER #1

Comment 1. *This paper is high interesting and important to the OA field, but warrant additional clarifications and further validation. For references to human cartilage among other papers this review is cited". Constitutive activities of estrogen-related receptors: transcriptional regulation of metabolism by the ERR pathways in health and disease" – which is a review not containing any patient material. For human studies, careful description of patients and materials must be included.*

Response: We originally cited this review to indicate diverse biological functions of ERRs under normal and pathophysiological conditions. In the revised manuscript, we substantially modified the second paragraph in the Introduction (p. 3 and 4). OA is currently considered to be a disease associated with metabolic disorders. Although this idea is still controversial, a number of studies have suggested an association of metabolic syndrome with OA pathogenesis. Because ERRs are known to regulate various metabolic processes, we focused on the functions of ERRs in regulating metabolic processes, citing the corresponding appropriate references. We additionally introduced common and isoform-specific biological functions of ERRs in this paragraph. The role of ERRs in cartilage biology is introduced in the third paragraph of the Introduction (p. 4).

Comment 2. *The use of very young mice in OA research is associated with many developmental effects. All the experiments done, would need to be reproduced in older animals, to support the impressive data seems in young animals.*

Response: This comment is essentially the same as Reviewers #3's Comment 5. We agree with the reviewer's concern and have addressed this issue by performing extensive additional experiments.

First, we have routinely used 10–12-week-old mice for DMM surgery and IA injection (Yang *et al.*, *Nat. Med.* 16, 687-693, 2010; Kim *et al.*, *Cell* 156, 730-743, 2014; Rhee *et al.*, *Ann. Rheum. Dis.* 76, 427-434, 2017; etc). In the current study, we used both 8- and 10-week-old mice. Because N numbers were greater for 8-week-old mice, we presented results obtained using 8-week-old mice in the original manuscript. During the revision process, we repeated experimental OA studies using 10-week-old mice, and present results of these experiment in the revised manuscript. We additionally addressed this issue as follows:

(1) We first analyzed 196 recently published papers that employed DMM methods (2013–2017). We found that a majority of such studies (71.9%, 141 papers) used young adult (8–11 weeks old) mice, and 25% (49 papers) used mature adult (12–39 weeks old) mice. Of papers that used young adult mice, 47 (39.8%) used 10-week-old mice for DMM surgery. The results of our

analysis are included in Supplementary Figure 4a and 4b in the revised manuscript and are described in the Results section (1st paragraph in p. 8).

- (2) DMM methods were originally developed in 10-week-old mice by Glasson et al. (Osteoarthritis Cartilage, 15, 1061-1069, 2007). Indeed, young animals have been shown to have a higher tissue-repair capacity and are thus better able to cope with tissue damage, leading to the recommendation that experimental OA studies should use 10-week-old mice (Poole et al., Osteoarthritis and Cartilage 18, S10-S16, 2010). Taking this literature background into consideration, we performed DMM surgery and IA injection of adenovirus using 8-, 10-, 12-, 16-, and 20-week-old mice. We found essentially the same pattern of OA development in all ages of mice examined. These results are included in Supplementary Figure 4c in the revised manuscript and are described in the Results section (1st paragraph in p. 8).
- (3) We also used various ages (8~20 weeks old) of mice for IA injection of adenoviruses. We saw a loss of glycosaminoglycans in articular cartilage in both juvenile (8-week-old) and young adult (10-week-old) mice (Supplementary Fig. 2b), suggesting that developmental stage is not a critical determinant of ERR γ -induced cartilage damage. Indeed, overexpression of HIF-2 α , which was previously shown to cause OA cartilage destruction, induced cartilage damage in mice at various ages from 8 to 20 weeks old (Supplementary Fig. 2c). These additional results are presented in Supplementary Figure 2b and 2c in the revised manuscript and are described in the Results section (1st paragraph in p. 7).
- (4) Because most studies have used 10-week-old mice for experimental OA studies, during the revision process, we repeated DMM surgeries and intra-articular (IA) injections using 10-week-old mice. We found essentially the same results with 8- and 10-week-old mice. We replaced original data obtained using 8-week-old mice with results from 10-week-old mice in all figures. Additional experiments performed during the revision processes also used 10-week-old mice (Figure 7 and Supplementary Figure 1). The corresponding text in Results and Methods sections has been modified accordingly.

Comment 3. *A substantial amount of estrogen receptor work has been undertaken in the bone field, many of these papers (in particular by Dr. Sims and Dr. Baron) should be discussed and cited.*

Response: In the revised manuscript, we discussed the relationship between estrogen receptors (ERs) and ERRs, and clearly indicated that they are not functionally related. The text now reads (2nd paragraph in p. 13): “The prevalence of OA is higher among women than men, and this prevalence increases after menopause. Various studies using animal models indicate that estrogen and estrogen signaling are associated with OA pathogenesis⁴⁸. For instance, mice lacking estrogen receptor- α develop more and larger osteophytes and a thinner lateral subchondral plate⁴⁹. A lack of estrogens

also increases subchondral bone remodeling⁵⁰. However, although the term “estrogen-related receptor (ERR)” reflects the molecular origin of the three family members, ERRs are functionally different from estrogen receptors. For instance, ERRs do not bind natural estrogens, nor are they directly involved in classic estrogen signaling pathways or biological processes^{18,24}. Furthermore, ERRs and estrogen receptors display strict binding site specificities⁵¹, with ERRs exhibiting a marked preference for the sequence TCAAGGTCA, an ERR response element (ERRE). Furthermore, unlike estrogen receptors, ERRs do not require ligand binding for their transcriptional activity^{18,24}. Therefore, the catabolic functions of ERR γ in OA pathogenesis might not be related to estrogen, even though estrogen is known to be associated with OA pathogenesis⁴⁸.”.

REVIEWER #2

In their manuscript Son et al. describe the induction of ERR γ expression in osteoarthritis (OA) and document the effect of this nuclear receptor on parameters of OA. Several mouse models of OA are used as well as several models of ERR γ modulation (transgenic and +/- animals, in vivo adenovirus-mediated overexpression or inhibition), rendering the conclusions convincing. Few things are known about OA and the authors' demonstration will be helpful in our understanding of the mechanisms of this disease as well as to propose ERR γ as a potential target to be deactivated in OA. The results of the experiments are clear and the paper is nicely written (although the Discussion section may deserved to be extended). Complementary experiments suggested below may reinforce the validity of the functional links proposed by the authors.

Response: We have extended our discussion in the revised manuscript to incorporate a consideration of the relevance of data obtained from our mouse models to human pathology, functions of ERR α and ERR γ in OA pathogenesis, phenotypes of our Tg mice and possible autoregulation of ERR γ , among other issues.

[Major points]

Comment 1. *The relationship between human pathology and mouse models in the present context is unclear. Indeed not only ERR γ but also ERR α is stimulated in the human, but not in mouse. ERR α and γ often display opposite effect (e.g. in EMT, Warburg effect, as prognosis factor in breast tumors); it may be that their effects on OA compensate each other's in human, but not mouse, questioning the validity of mouse models for studying ERRs' effect in OA, a point that should be discussed. The authors should show the basal relative expressions of ERR α and γ (at the mRNA and protein levels), in damaged and undamaged human OA cartilage (i.e. it may be that ERR α expression is negligible relative to ERR γ , ruining my above concern). Also: is the same discrepancy observed when comparing human to mouse cartilage cells in vitro (upon IL-6 exposure for instance)? Whatever the answer to these questions, the relevance of the present data to human pathology should be more discussed.*

Response: We appreciate the reviewer's comment. In response, we have evaluated mRNA levels of ERRs by calculating their expression levels relative to GAPDH mRNA levels in OA cartilage of humans (damaged and undamaged) and mice (sham-treated and DMM). We found that, although all ERR isoforms in damaged regions of human OA showed significant fold-increases compared with undamaged regions, mRNA levels of ERR α and β in OA cartilage of both humans and mice were negligible relative to those of ERR γ . Therefore, in both human and mouse OA cartilage, ERR γ is the most abundantly upregulated ERR isoform, clearly indicating the importance of ERR γ in OA pathogenesis. These additional data are included in Figure 1a and 1b of the revised manuscript and are described in the Results section (1st paragraph in p. 5).

Notwithstanding the above results, we also evaluated $ERR\alpha$ protein by IHC in human and mouse OA cartilage, and found no detectable expression. These results are consistent with mRNA levels of $ERR\alpha$ in human and mouse OA cartilage. Additionally, whereas IL-6 treatment stimulated $ERR\gamma$ expression in chondrocytes, it did not induce expression of $ERR\alpha$ or $ERR\beta$. These additional results are included in Supplementary Figure 1 of the revised manuscript and are described in the Results section (1st paragraph in p. 5). We also IA-injected Ad-*Esrra* to overexpress $ERR\alpha$ in joints tissues. Although $ERR\alpha$ was efficiently overexpressed in cartilage, we found no evidence of cartilage damage or synovial inflammation, clearly indicating that $ERR\gamma$, but not $ERR\alpha$, regulates OA pathogenesis. These results are presented in Supplementary Figure 7 and are described in the corresponding Results section (3rd paragraph in p. 10).

We also discussed the relevance of data obtained from our mouse model to human pathology and functions of $ERR\alpha$ and $ERR\gamma$ in OA pathogenesis in the last paragraph of the Discussion section (p. 14~15).

Comment 2. *There are missing links in the authors' demonstration of an IL-1 β /IL-6/ $ERR\gamma$ axis (Fig. 2). Authors could show that:*

(1) IL-6 protein is indeed secreted upon IL-1 β exposure in their system.

Response: We have previously shown that IL-1 β stimulates IL-6 production in chondrocytes (Ryu et al., *Arthritis Rheum.* 63, 2732-43, 2011). We further verified IL-1 β -induced IL-6 expression by qRT-PCR analysis, demonstrating that IL-6 mRNA is increased in chondrocytes treated with IL-1 β (and also by overexpression of HIF-2 α and ZIP8); these results are presented in Figure 2a of the revised manuscript. We also demonstrated that IL-1 β increased the amount of extracellular secreted IL-6 protein in a concentration-dependent manner (Figure 2c). Finally, we detected cellular IL-6 in IL-1 β -treated chondrocytes by immunofluorescence microscopy (Figure 2b). These additional data are presented in Figure 2a–2c and are described in text of the corresponding Results section (p. 5~6).

(2) Blocking IL-6 impairs $ERR\gamma$ increased expression upon IL-1 β .

Response: To address this issue, we first demonstrated IL-1 β -induced $ERR\gamma$ expression by qRT-PCR analysis; these additional data are presented in Figure 2f (left panel) of the revised manuscript. We then confirmed that siRNA-mediated knockdown of IL-6 abrogated IL-1 β -induced $ERR\gamma$ expression. These data are now included in Figure 2f (middle panel) of the revised manuscript and are described in the text of the corresponding Results section (1st paragraph in p. 6).

(3) IL-1 β treatment increases ERR γ expression in their system. This data should be available in their microarray study, but could be confirmed by qPCR; by the way what is the status of ERR γ and β expression (and potential variations) in their microarray.

Response: As described above, we presented results demonstrating IL-1 β –induced ERR γ expression in chondrocytes in Figure 2f (left panel) of the revised manuscript.

Regarding microarray analysis, we have added results of ERR isoform expression in chondrocytes treated with IL-1 β or infected with Ad-*Epas1* to overexpress HIF-2 α or Ad-*Slc39a8* to overexpress ZIP8. These results are now presented in Supplementary Table 1 of the revised manuscript. Interestingly, whereas our qRT-PCR analysis indicated induction of ERR γ by IL-1 β , our microarray analysis revealed no significant difference in the expression of ERR γ as well as ERR α and - β . We postulate that these differences are attributable to the duration of IL-1 β treatment. We found that IL-1 β –induced ERR γ expression peaked at 12 hours and decreased thereafter (Figure 2e). The fact that we treated with IL-1 β for 36 hours in microarray analyses could account for the absence of detectable ERR γ expression in these samples. This issue is now considered in the Results section of our revised manuscript (p. 5~p. 6).

Regarding ERR variants, there are two variants of mouse ERR β and two of ERR γ . The Affymetrix Gene Chip mouse gene 2.0 ST array that we used does not contain these ERR variants.

(4) Is the effect of IL-6 on ERR γ also JAK2/STAT3-dependent as in the liver? Authors could use the AG490 JAK2 inhibitor to show this.

Response: As suggested by the reviewer, we used the JAK2 inhibitor AG490 to examine the role of JAK2/STAT3 signaling in IL-6–induced ERR γ expression. We found that AG490 blocked IL-6–induced ERR γ expression. These additional data are presented in Figure 2f (right panel) and are described in the Results section (1st paragraph in p. 6).

(5) The time point of maximal ERR γ induction by IL-6 seems discrepant between the two panels of Fig 2b. Any comment?

Response: During the revision process, we extensively characterized IL-6–induced ERR γ expression, examining the time- and concentration-dependence of IL-6 induction of ERR γ mRNA. We found that IL-6–induced ERR γ expression peaked at 12 hours in a concentration-dependent manner. These

additional data are included in Figure 2e and are described in the text in the corresponding Results section (1st paragraph in p. 6).

(6) GSK5182 reduces IL6-dependent ERR γ increased expression (Fig 7f). Is ERR γ autoregulated at the transcriptional level?

Response: We originally found that inhibition of ERR γ with GSK5182 reduced mRNA levels of ERR γ , suggesting possible autoregulation of ERR γ . Indeed, we previously demonstrated that ERR γ protein levels are decreased by GSK5182 treatment (ref. #42), possibly reflecting the operation of an autoregulatory feed-forward mechanism that regulates ERR γ gene expression (ref. #43). We additionally found that IL-1 β -induced upregulation of ERR α in chondrocytes is reduced by the ERR α inverse agonist XCT790 (ref. #32). Therefore, expression of both ERR α and ERR γ in chondrocytes appears to be regulated by positive feedback mechanisms. This is discussed on p. 12 (1st paragraph) of the Discussion section. We further discussed amplification of catabolic signaling during OA pathogenesis on p. 12 (1st paragraph) of the Discussion.

(7) Besides, IL-1 β elicits a much stronger increase in IL-6 expression than do HIF-2 α or ZIP8 overexpression, at least at the microarray. Is this confirmed by independent qPCR and at the level of secreted IL-6 protein? If yes, would the lower amount of IL-6 produced upon HIF-2 α or ZIP8 overexpression be sufficient to induce ERR γ (as well as MMP3/MMP13)?

Response: To address this issue, we performed qRT-PCR analyses, and found that IL-1 β upregulated IL-6 by up to ~900-fold, whereas HIF-2 α and ZIP8 upregulated IL-6 by up to ~20- and ~40-fold, respectively. These results are included in Figure 2a and are described in the Results section of the revised manuscript (p. 6, 1st paragraph).

Although IL-1 β upregulates ERR γ , neither HIF-2 α nor ZIP8 overexpression induced a marked upregulation of ERR γ , probably owing to the relatively small degree of ERR γ upregulation. However, we previously demonstrated that HIF-2 α directly targets and induces expression of the genes encoding MMP3 and MMP13 (Yang et al., Nat. Med. 16, 687-693, 2010) and that ZIP8 promotes MMP3/MMP13 expression through activation of the transcription factor MTF1 (Kim et al., Cell. 156, 730-743, 2014).

Comment-3. *There are missing links in the relationships with MMP3/MMP13.*

(1) Effect of IL-6 on MMP3/13 should be shown at the protein level (Fig. 2)

Response: During the revision process, we extensively characterized the relationships among IL-6, ERR γ , and MMP3/MMP13. We first demonstrated the time- and concentration-dependence of IL-6–induced expression of MMP3 and MMP13 mRNAs. Using Western blotting, we also demonstrated an increase in the levels of extracellular MMP3 and MMP13 proteins secreted by IL-6–treated chondrocytes. These additional data are included in Figure 2d and 2e and are described in the corresponding text (p. 6, 1st paragraph).

We additionally demonstrated IL-1 β –induced production of IL-6 in chondrocytes by Western blotting and immunofluorescence microscopy (Figure 2b and 2c).

(2) In their experiments (Fig. 7b), do the authors examine secreted/processed MMP3/MMP13 proteins (that is, examining culture medium instead of cellular extracts)?

Response: In the Western blots shown in the original Figure 7b (Figure 7a in the revised manuscript), MMP3 and MMP13 were detected in conditioned medium of chondrocyte cultures. Therefore, the detected MMPs represent secreted and processed MMP3 and MMP13. This is now indicated in the Results section of the revised manuscript (p. 9, 2nd paragraph).

(3) What is the status of MMP3/13 in ERR γ Tg and ERR γ ^{+/-} animals (when induced by IL-1 β or IL-6 for cultured chondrocytes or in DMM challenged animals)

Response: We detected MMP3 and MMP13 by IHC in cartilage sections of sham- and DMM-operated WT, Tg, and *Esrrg*^{+/-} mice. Compared with sham-operated mice, the cartilage of DMM-operated WT mice exhibited increased expression of MMP3 and MMP13 proteins in the damaged regions of cartilage, consistent with our previous report. The increases in the expression of MMP3 and MMP13 proteins in DMM-operated mice were further enhanced in DMM-operated Tg mice. In contrast, DMM-induced upregulation of MMP3 and MMP13 proteins was markedly abrogated in *Esrrg*^{+/-} mice. These results are now included in Supplementary Figure 3d (for Tg mice) and Supplementary Figure 5d (for *Esrrg*^{+/-} mice), and are described in the corresponding Results section of the revised manuscript (p. 7, last paragraph; p. 8, 2nd paragraphs).

[Minor points]

(1) The reviews (ref 15, 24, 25) are not appropriate to reference the fact that IL-6 stimulates ERR γ expression. The Kim paper (ref 30) is more appropriate.

Response: In the revised manuscript, we replaced these references with a citation of our recent review that describes various membrane receptors and cellular stresses that induce ERR γ expression (Misra et al., ERR γ : a junior orphan with a senior role in metabolism. *Trend Endocrinol. Metab.* 28, 261- 272, 2017).

(2) A comment is lacking at the end of first chapter in Results section to explain why the authors examine MMP3 and MMP13 in Fig 2b. Besides this result is not commented per se.

Response: We examined MMP3 and MMP13 because their corresponding genes are targets of IL-6 in chondrocytes (Ryu et al., *Arthritis Rheum.* 63, 2732-43, 2011). This is now indicated and described in the revised manuscript (p. 6, 1st paragraph).

We additionally characterized IL-1 β -induced IL-6 expression and IL-6-induced expression of ERR γ , MMP3 and MMP13, including concentration- and time-dependence and production of secreted MMP3 and MMP13. These additional data are included in Figure 2b~2e, and are described in the corresponding Results section (p. 6, 1st paragraph) of the revised manuscript.

(3) The difference between WT and Tg animals after DMM (fig 4d) could be better illustrated: show what we have to look at with arrows for example.

Response: As suggested by the reviewer, we indicated cartilage (c), subchondral bone (b), and osteophyte (o) in the images of joint sections in Figures 3c, 4d, 5c, and 6c of our revised manuscript.

(4) Fig 5 should not be labeled with "KO" since heterozygote animals are used (+/- would be more appropriate)

Response: We have corrected the text as suggested by the reviewer.

(5) Cardelli et al (2013) reported no effect on Sox9 and increased Col2 expression in ERR γ Tg animals. This discrepancy could be discussed.

Response: We share the reviewer's concern and thank the reviewer for this comment. Cardelli et al. (2013) reported that cartilage-specific ERR γ Tg mice exhibit chondrodysplasia, in which repressed

chondrocyte proliferation decreases the proliferative zone of the growth plate and reduces bone length.

We found no differences in body size between ERR γ Tg mice and WT littermates and showed skeletal staining of E18.5 whole embryos in the original manuscript. The issue raised by the reviewer was addressed during revision by performing an additional analysis of our ERR γ Tg mice. This analysis included alcian blue staining of 2-week-old metatarsal bones in Tg and WT littermates and measurement of the length of resting/proliferative and hypertrophic zones of growth plate. Compared with WT littermates, our Tg mice exhibited a similar whole-body skeletal structure, metatarsal bone length, and length of proliferative and hypertrophic zones. Therefore, the phenotypes of our Tg mice are different from those of Tg mice generated by Cardelli *et al.* (2013). Although it is not clear what accounts for these differences, they may be attributable to differences in expression systems and the expression levels of target genes. We extensively discuss this issue in the Discussion section of the revised manuscript (p. 12~13). Additional results are included in Supplementary Figure 3b and 3c and are described in the corresponding Results section (p. 7, 2nd paragraph).

Unlike the elevated expression of type II collagen observed in ERR γ Tg mice generated by Cardelli *et al.* (2013), we found that overexpression of ERR γ (via infection with Ad-*Esrrg*) in primary cultured WT chondrocytes caused a slight, but significant, downregulation of SOX9, type II collagen, and aggrecan. Again, these observed differences may be related to differences in expression systems and levels of ERR γ expression. We address differences between these two Tg mouse lines in the Discussion section of the revised manuscript (p. 12~13). Additional data are also presented in Supplementary Figure 3b and 3c and are described in the Results section (p. 7, 2nd paragraph).

(6) First sentence in Discussion section is too straightforward. i- authors do not distinguish between the different ERR; ii- muscular effects of ERRs were only shown in cell culture systems (as far as I know); ii- ERR effects in bone were shown in mouse models and are unclear in the human. Discussion section is really short and could deserve extension.

Response: We thank the reviewer for this comment. In response, we have substantially modified our Discussion. Additionally, we extended the discussion of our findings to include a consideration of the relevance of data obtained from our mouse model to human pathology, functions of ERR α and ERR γ in OA pathogenesis, phenotypes of our Tg mice, and possible feedback regulation of ERR γ expression.

REVIEWER #3

In this manuscript, the authors examined the role of the orphan nuclear receptors estrogen-related receptors (ERRs) in osteoarthritis (OA) pathogenesis. Using both gain-of-function and loss-of-function approaches, the authors provide ample evidence to suggest ERR γ contributes to OA pathogenesis, in part through catabolic regulation of MMP13 and MMP3 expression. As the roles of ERRs in OA remain previously unexplored, this finding gives good insight into the understanding of nuclear receptor function in OA and suggests ERR γ to be a novel therapeutic target. However, certain fundamental concerns exist regarding the presentation and discussion of the results that should be addressed.

Comment 1. *In the introduction, the authors refer to findings from reference #22 where cartilage-specific over-expression of ERR γ has previously been shown to result in chondrodysplasia and reduced chondrocyte proliferation. This contradicts the results in Figure 3 showing no defects in skeletal development in the cartilage-specific over-expressors generated here. Further studies should be conducted at multiple time-points (instead of just at E18.5) throughout development to confirm ERR γ overexpression to have no effect on skeletal development and chondrocyte differentiation, as these effects could potentially alter the conclusions made from these animals in the context of OA.*

Response: We share the reviewer's concern and thank the reviewer for this comment. Cardelli et al. (2013) reported that cartilage-specific ERR γ Tg mice exhibit chondrodysplasia, in which repressed chondrocyte proliferation decreases the proliferative zone of the growth plate and reduces bone length.

We found no differences in body size between ERR γ Tg mice and WT littermates and showed skeletal staining of E18.5 whole embryos in the original manuscript. The issue raised by the reviewer was addressed during revision by performing an additional analysis of our ERR γ Tg mice. This analysis included alcian blue staining of 2-week-old metatarsal bones in Tg and WT littermates and measurement of the length of resting/proliferative and hypertrophic zones of growth plate. Compared with WT littermates, our Tg mice exhibited a similar whole-body skeletal structure, metatarsal bone length, and length of proliferative and hypertrophic zones. Therefore, the phenotypes of our Tg mice are different from those of Tg mice generated by Cardelli et al. (2013). Although it is not clear what accounts for these differences, they may be attributable to differences in expression systems and the expression levels of target genes. We extensively discuss this issue in the Discussion section of the revised manuscript (p. 12~13). Additional results are included in Supplementary Figure 3b and 3c and are described in the corresponding Results section (p. 7, 2nd paragraph).

Comment 2. *The authors should consider showing more representative images of ERRy IHC from Ad-C and Ad-Esrrg sections, as the degree of adenovirus penetration into the articular cartilage is not clear from some images (e.g. Fig. 3a only shows the meniscus and very superficial layers of the cartilage and not deeper layers of cartilage and the subchondral bone).*

Response: We replaced original Figure 3a to show more representative images of ERRy IHC results from mice IA-injected with Ad-Esrrg. In the revised manuscript, Figure 3a is an image of ERRy IHC in a cartilage section of a 10-week-old mouse IA-injected with Ad-Esrrg. ERRy IHC of the meniscus and synovium is presented separately in Supplementary Figure 2a. In addition to changes in Figure 3a, we also replaced the original ERRy IHC with more representative images in the revised figures. These include Figures 4b, 5a and 6a, and Supplementary Figures 2a, 3a and 5b.

Comment 3. *It is not clear whether the OARSI scoring used was modified in any way, as certain scores seem higher than expected compared to their corresponding histological images (e.g. Fig 5f, 5g show only proteoglycan loss and no actual structural damage which would result in a much lower score). Please indicate any scoring modifications in the methods section or provide more representative histological images to avoid confusion. More generally some of the OA phenotypes mentioned appear rather mild, consisting largely of proteoglycan depletion without damage to the cartilage surface.*

Response: We used a previously reported semi-quantitative OARSI scoring method (Glasson et al., Osteoarthritis and Cartilage. 18 Suppl 3, S17-23, 2010) without modification. The severity of cartilage destruction was expressed as maximal scores, which were combined for the entire joint: medial femoral condyle, medial tibial plateau, lateral femoral condyle, lateral tibial plateau. The scoring system is as follows:

Score 0: normal cartilage.

Score 0.5: loss of proteoglycan staining without structural changes to the articular surface.

Score 1: superficial fibrillation without loss of cartilage.

Score 2: vertical clefts down to the layer immediately below the superficial layer and some loss of surface lamina.

Score 3: vertical clefts/erosion to the calcified cartilage extending to <25% of the articular surface.

Score 4: vertical clefts/erosion to the calcified cartilage extending to 25–50% of the articular surface.

Score 5: vertical clefts/erosion to the calcified cartilage extending to 50–75% of the articular surface.

Score 6: vertical clefts/erosion to the calcified cartilage extending to >75% of the articular surface.

To address this issue, we performed DMM surgery and IA injection using 10-week-old mice (as

suggested by Reviewers 1 and 3) rather than 8-week-old mice; OARSI grade was scored by five independent observers in a blinded manner. The revised figures include Figures 3d, 4c, 5b, 6b and 7f, and Supplementary Figure 4c. We also replaced bar graphs with column scatter plots, which provide more information. Additionally, we excluded scoring OARSI grade (in Figure 1c of the revised manuscript) in mice IA-injected with Ad-*Epas1* or Ad-*Slc39a8*, but described the loss of glycosaminoglycan in articular cartilage above the tidemark in these mice. OARSI grade scoring methods are described in the Method section of the revised manuscript (p. 16).

Comment 4. *Please provide more representative histological images that show the whole joint in OA studies. Often the images are very cropped and only show one area of the cartilage, or only show the femur and not both the femur and tibia (e.g. Fig 4d vs Fig 5d). Please be consistent in presenting histological data to avoid bias.*

Response: We thank the reviewer for this observation. As suggested by the reviewer, we replaced original images with more representative histological images. These include Figures 3c, 4d, 5c, and 6c. We also present additional data with these criteria, including Figure 7f and Supplementary Figure 4c, in the revised manuscript.

Comment 5. *It is not clear why 8-week old animals were chosen for OA induction through DMM surgery or IA injections. Please discuss thoroughly the rationale behind using animals this young and why only one time-point post-surgery/post-injection was examined. 8 week-old mice are not mature skeletally and therefore suboptimal for these studies.*

Response: We agree with the reviewer's concern, and addressed this issue by performing extensive additional experiments, as described in our response to Reviewer 1, who raised essentially the same point. We have addressed this issue by performing extensive additional experiments.

First, we have routinely used 10–12-week-old mice for DMM surgery and IA injection (Yang *et al.*, *Nat. Med.* 16, 687-693, 2010; Kim *et al.*, *Cell* 156, 730-743, 2014; Rhee *et al.*, *Ann. Rheum. Dis.* 76, 427-434, 2017; etc). In the current study, we used both 8- and 10-week-old mice. Because N numbers were greater for 8-week-old mice, we presented results obtained using 8-week-old mice in the original manuscript. During the revision process, we repeated experimental OA studies using 10-week-old mice, and present results of these experiment in the revised manuscript. We additionally addressed this issue as follows:

(1) We first analyzed 196 recently published papers that employed DMM methods (2013–2017). We found that a majority of such studies (71.9%, 141 papers) used young adult (8–11 weeks old)

mice, and 25% (49 papers) used mature adult (12–39 weeks old) mice. Of papers that used young adult mice, 47 (39.8%) used 10-week-old mice for DMM surgery. The results of our analysis are included in Supplementary Figure 4a and 4b in the revised manuscript and are described in the Results section (1st paragraph in p. 8).

- (2) DMM methods were originally developed in 10-week-old mice by Glasson et al. (Osteoarthritis Cartilage, 15, 1061-1069, 2007). Indeed, young animals have been shown to have a higher tissue-repair capacity and are thus better able to cope with tissue damage, leading to the recommendation that experimental OA studies should use 10-week-old mice (Poole et al., Osteoarthritis and Cartilage 18, S10-S16, 2010). Taking this literature background into consideration, we performed DMM surgery and IA injection of adenovirus using 8-, 10-, 12-, 16-, and 20-week-old mice. We found essentially the same pattern of OA development in all ages of mice examined. These results are included in Supplementary Figure 4c in the revised manuscript and are described in the Results section (1st paragraph in p. 8).
- (3) We also used various ages (8~20 weeks old) of mice for IA injection of adenoviruses. We saw a loss of glycosaminoglycans in articular cartilage in both juvenile (8-week-old) and young adult (10-week-old) mice (Supplementary Fig. 2b), suggesting that developmental stage is not a critical determinant of ERR γ -induced cartilage damage. Indeed, overexpression of HIF-2 α , which was previously shown to cause OA cartilage destruction, induced cartilage damage in mice at various ages from 8 to 20 weeks old (Supplementary Fig. 2c). These additional results are presented in Supplementary Figure 2b and 2c in the revised manuscript and are described in the Results section (1st paragraph in p. 7).
- (4) Because most studies have used 10-week-old mice for experimental OA studies, during the revision process, we repeated DMM surgeries and intra-articular (IA) injections using 10-week-old mice. We found essentially the same results with 8- and 10-week-old mice. We replaced original data obtained using 8-week-old mice with results from 10-week-old mice in all figures. Additional experiments performed during the revision processes also used 10-week-old mice (Figure 7 and Supplementary Figure 1). The corresponding text in Results and Methods sections has been modified accordingly.

Comment-6. *The authors should provide an explanation for why the inverse agonist (GSK5182), instead of shRNA, was used in vitro (Fig. 7f), and why this inverse agonist caused decreased mRNA levels of ERR γ . The authors should consider conducting further studies to examine potential feedback regulation of ERR γ expression, as well as validating the effect of GSK5182 in vivo.*

Response: We used the inverse agonist GSK5182 to examine whether inhibition of ERR γ transcriptional activity (instead of its expression) is necessary for the upregulation of MMP3 and MMP13. We additionally thought that this might be an approach for examining whether GSK5182

could be used as a potential drug to treat OA pathogenesis, as suggested by the reviewer. Indeed, during the revision process, we tested the effects of GSK5182 in OA pathogenesis caused by DMM surgery. We found that intraperitoneal administration of GSK5182 significantly inhibited DMM-induced cartilage destruction, osteophyte development, and subchondral bone sclerosis. These results are included in Figure 7f of the revised manuscript and are described in the Results section (p. 10, 2nd paragraph). We also discussed this finding in the last paragraph of the Discussion section (p. 14).

We originally found that inhibition of ERR γ with GSK5182 reduced mRNA levels of ERR γ , suggesting possible autoregulation of ERR γ . Indeed, we previously demonstrated that ERR γ protein levels are decreased by GSK5182 treatment (ref. #43), an effect that might be attributable to an autoregulatory feed-forward mechanism in the regulation of ERR γ gene expression (ref. #44). We additionally found that IL-1 β -induced upregulation of ERR α in chondrocytes is reduced by the ERR α inverse agonist XCT790 (ref. #32). Therefore, expression of both ERR α and ERR γ in chondrocytes appears to be regulated by positive feedback mechanisms. This is discussed on p. 12 (1st paragraph) of the Discussion section, and amplification of catabolic signaling during OA pathogenesis is further discussed on p. 12 (1st paragraph) of the Discussion.

Comment-7. *Please perform quantitative real-time PCR and quantitative ChIP analysis. Changes in mRNA levels of Adamts4, Adamts5, Sox9 and Col2a1 may not be detected through qualitative PCR, and could be the reason behind the observed decrease in protein levels of Sox9 and Col2a1.*

Response: As suggested by the reviewer, we performed quantitative ChIP analyses; the results are included in Supplementary Figure 6c of the revised manuscript.

We also performed qRT-PCR analyses of ADAMTS4, ADAMTS5, SOX9, Coll-II, and aggrecan. We found that overexpression of ERR γ downregulated expression of SOX9, Coll-II and aggrecan, although the degree of downregulation was small. Expression of ADAMTS4 and ADAMTS5 was not modulated by ERR γ overexpression. These additional data are presented in Supplementary Figure 6b and are described in the corresponding Results section of the revised manuscript (p. 9).

REVIEWERS' COMMENTS:

Reviewer #1 (Remarks to the Author):

The Authors have improved the manuscript to an extent that this reviewer did not think was possible. Congratulations on an important study.

Reviewer #2 (Remarks to the Author):

All comments I raised have been satisfactorily addressed. I do not have further concerns.

Reviewer #3 (Remarks to the Author):

The manuscript is quite improved through the addition of new experiments, changes to figures, and editorial changes.

However a number of issues remain:

- 1) some histological images show a larger part of the joint, but some are still very cropped and restricted to the tibia. Examples include most immunohistochemistry, all histology in fig. 1, fig 7f, suppl fig 3 etc.
- 2) the qChIP data should be included in the main manuscript instead of the ChIP data since they are more meaningful

Reviewers' comments:

Reviewer #1 (Remarks to the Author):

The authors have improved the manuscript to an extent that this reviewer did not think was possible. Congratulations on an important study.

Reviewer #2 (Remarks to the Author):

All comments I raised have been satisfactorily addressed. I do not have further concerns.

Reviewer #3 (Remarks to the Author):

The manuscript is quite improved through the addition of new experiments, changes to figures, and editorial changes. However a number of issues remain:

- 1) Some histological images show a larger part of the joint, but some are still very cropped and restricted to the tibia. Examples include most immunohistochemistry, all histology in fig. 1, fig 7f, suppl fig 3 etc.
- 2) The qChIP data should be included in the main manuscript instead of the ChIP data since they are more meaningful.

Point-to-point response to reviewers' comments

Reviewer #3

Comment 1. *Some histological images show a larger part of the joint, but some are still very cropped and restricted to the tibia. Examples include most immunohistochemistry, all histology in fig. 1, fig 7f, suppl fig 3 etc.*

Response: As suggested by the reviewer, we added whole-joint images for histology and immunohistochemistry. These include:

Figures 1f (original Figure 1b)

Figure 7f

Supplementary Figure 1a and 1b (whole-joint images of original Figure 1c)

Supplementary Figure 2b (original supplementary Figure 1b)

Supplementary Figure 3c (original supplementary Figure 2c)

Supplementary Figure 4d (original supplementary Figure 3d)

Supplementary Figure 8a (original supplementary Figure 7a)

We did not include whole-joint images for supplementary Figure 6d and 6e because the figures are crowded and whole-joint images are presented in other figures.

Supplementary Figure 1 in the revised manuscript was additionally added during revision process to address this issue. Figure 1c in the original manuscript was moved to Supplementary Figure 1.

Comment 2. *The qChIP data should be included in the main manuscript instead of the ChIP data since they are more meaningful.*

Response: As suggested by the reviewer, qChIP data are included in Figure 7d, whereas ChIP data are moved to Supplementary Figure 7c.